# Improving Safe Offline Reinforcement Learning via Dual-Guide Diffuser

## Abstract

In offline safe reinforcement learning (OSRL), upholding safety guarantees while optimizing task performance remains a significant challenge, particularly when dealing with adaptive and time-varying constraints. While recent approaches based on actor-critic methods or generative models have made progress, they often struggle with robust safety adherence across diverse and dynamic conditions. For diffusion models specifically, a key bottleneck is the reliance on unreliable cost classifiers for safety guidance. To address these limitations, we propose **SDGD** (*Safe Dual-Guide Diffuser*), a novel framework that decouples safety and performance optimization. SDGD leverages classifier-free guidance (CFG) to strictly enforce cost constraints while simultaneously using classifier guidance (CG) to steer generation towards high-reward outcomes. This dual-guide mechanism robustly handles cost limits that change dynamically within a single episode. We provide error bounds of the estimation on reward and cost, offering performance and safety guarantees. Extensive evaluations on the DSRL benchmark demonstrate that SDGD establishes a new state-of-the-art, achieving safety in 94.7% of tasks (36/38). Crucially, our method extends the aggregate Pareto frontier in the reward-cost space, achieving a superior trade-off in the safe region.

## 1 Introduction

Balancing task performance with stringent safety guarantees is a fundamental challenge in reinforcement learning, especially for real-world applications like autonomous navigation Duan et al. (2024); Peng et al. (2024); Zheng et al. (2025); He et al. (2024) and robotics Gao et al. (2024); Hoang et al. (2025); Hung et al. (2025), where constraint violations can have severe consequences. While online safe RL methods Yang et al. (2023); Achiam et al. (2017); Yang et al. (2020); Zhang et al. (2020) offer potential solutions, their need for extensive and potentially unsafe environmental interaction during training makes them risky for direct real-world deployment. Consequently, offline safe RL (OSRL) Guo et al. (2025b); Zheng et al. (2024); Xu et al. (2022); Liu et al. (2023b) has emerged as a more viable paradigm. OSRL aims to learn safe policies entirely from pre-collected datasets, thereby avoiding hazardous online interactions, making it better suited for safety-critical applications.

A significant limitation of many early OSRL methods Zheng et al. (2024); Xu et al. (2022); Lee et al. (2022) is that they are designed for a fixed cost limit $l$, requiring costly retraining to adapt to new safety requirements. To address this, recent actor-critic approaches have emerged. Methods like CCAC Guo et al. (2025a) learn a single policy conditioned on the cost limit, while CAPS Chemingui et al. (2025) trains multiple specialized policies and switches between them at runtime. While effective for adaptation, these methods rely on step-by-step action selection, which in offline settings can suffer from error accumulation and limited temporal abstraction Chen et al. (2024).

An alternative paradigm that directly tackles adaptive cost limits is using generative models. However, autoregressive (AR) models like CDT Liu et al. (2023b) generate trajectories sequentially. This process inherits the myopia of step-by-step methods and is thus prone to compounding errors over long horizons, making it challenging to reliably satisfy complex constraints as illustrated in Figure 1 (left). In contrast, diffusion models generate the entire trajectory simultaneously, a holistic process that avoids the compounding errors inherent to sequential generation. Despite this architectural advantage, the practical effectiveness of current diffusion-based methods Lin et al. (2023) is crippled by a reliance on an unreliable cost classifier $p_\phi(\tau_t)$, which struggles to estimate cumulative costs

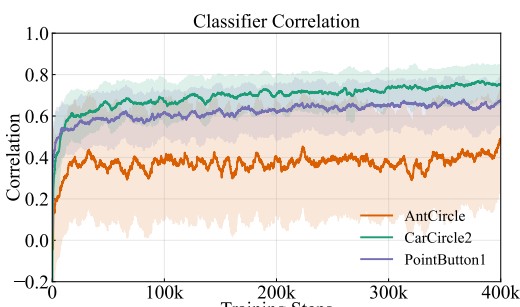

Figure 1: A comparison of generative models handling a time-varying cost limit. SDGD succeeds, while Autoregression struggles with compounding errors, and Classifier Guidance Diffusion fails due to an unreliable safety classifier.

$\hat{C}$ from noisy trajectories. As our analysis shows (Figure 2), the correlation between a classifier's prediction and the true cost is often low and unstable, leading to catastrophic safety failures as depicted in Figure 1 (middle). This exposes a critical bottleneck that our work aims to solve.

To overcome these limitations, we propose **SDGD** (*Safe Dual-Guide Diffuser*), a diffusion-based framework that provides accurate guidance by decoupling safety and performance. Our core insight is to apply a dual-guide mechanism to a Sparse-States Dense-Actions trajectory representation: we use classifier-free guidance (CFG) Ho & Salimans (2022) to strictly enforce cost constraints, while simultaneously employing classifier guidance (CG) Dhariwal & Nichol (2021) to steer the trajectory towards high-reward outcomes.

Our contributions are: (1) We propose SDGD, a novel dual-guide diffusion framework that decouples safety and performance by synergizing CFG for cost-conditioned trajectory generation and CG for reward optimization. (2) We provide a theoretical analysis that establishes formal guarantees for both safety and near-optimality, explicitly bounding the performance gap caused by the distribution shift inherent to the offline setting. (3) Extensive evaluations on 38 DSRL benchmark tasks demonstrate that SDGD surpasses existing methods in safety and performance.

Figure 2: Unreliability of Cost Classifiers for Safety Guidance. Pearson correlation between a classifier's predicted and true cumulative cost during training on three DSRL tasks. The low and unstable correlation highlights the unreliability of this classifier-based guidance.

## 2 PRELIMINARIES

### 2.1 SAFE REINFORCEMENT LEARNING

Safe Reinforcement Learning problems are commonly formulated using Constrained Markov Decision Process (CMDP) Ames et al. (2019). A CMDP is defined by the tuple $\mathcal{M} = (\mathcal{S}, \mathcal{A}, T, r, c, l, \gamma)$, where $\mathcal{S}$ represents the state space; $\mathcal{A}$ denotes the action space; $T = \mathcal{S} \times \mathcal{A} \rightarrow \Delta(\mathcal{S})$ describes the transition dynamics; $r = \mathcal{S} \times \mathcal{A} \rightarrow [R_{\min}, R_{\max}]$ is the state-action reward function; $c = \mathcal{S} \times \mathcal{A} \rightarrow [0, C_{\max}]$ is the state-action cost function; $l$ is a cost limit and $\gamma \in (0, 1]$ is the discount factor. The finite trajectory $\tau = \{s_i, a_i, ..., s_{i+L}, a_{i+L}\}$, where $L$ is the maximum episode length. The total discounted reward and cost for a trajectory $\tau$ are defined as $R(\tau) = \sum_{t=i}^{i+L} \gamma^{t-i} r(s_t, a_t)$ and $C(\tau) = \sum_{t=i}^{i+L} \gamma^{t-i} c(s_t, a_t)$, respectively. The core objective combines reward maximization with time-varying safety constraints:

$$\max_{\pi} \mathbb{E}_{\tau \sim \pi}[R(\tau)] \quad \text{s.t.} \quad \mathbb{E}_{\tau \sim \pi}[C_k(\tau)] \leq l_k, \quad \forall k \in 0, ..., K. \tag{1}$$

Here the $\mathbb{E}_{\tau \sim \pi}[\cdot]$ denotes the expectation over trajectories $\tau$ generated by the policy $\pi$. Each $C_k(\tau) = \sum_{t=k}^{k+T_k} \gamma^{t-k} c(s_t, a_t)$ is the cost of sub-trajectories with predefined horizon $T_k$, constrained by its corresponding limit $l_k$. This general objective naturally covers fixed limits ($K = 0$) and time-varying limits ($K > 0$) through segmented cost limits.

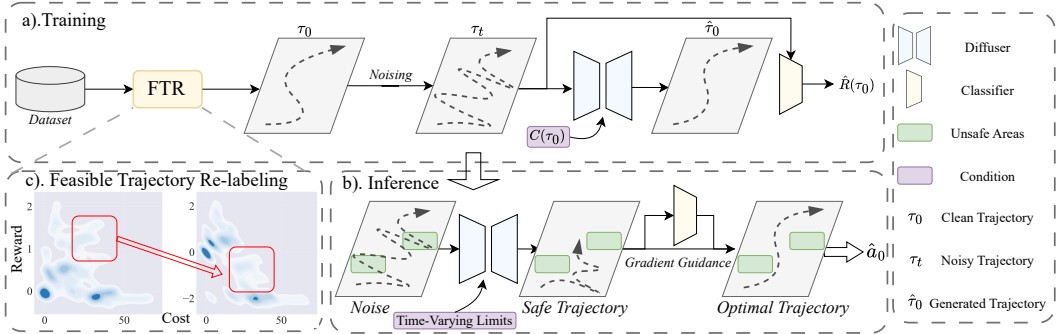

**Figure 3: The framework of SDGD.** (a) represents the training process. (b) represents the inference process with real-time constraints. (c) compares the cumulative reward cost distribution of original (left) and *FTR* processed (right) trajectories. *FTR* suppresses high-reward but high-cost trajectories (red block), preventing reward gradients from driving trajectories toward high-cost regions.

## 2.2 DIFFUSION MODELS

Diffusion models Ho et al. (2020); Nichol & Dhariwal (2021) are generative models that learn a data distribution by reversing a fixed process that gradually adds noise to data. The noising process transforms a clean sample $x_0$ into a noisy sample $x_t$ over $N$ steps, where $x_t$ can be sampled directly via $x_0$ using $q(x_t|x_0) = \mathcal{N}(x_t; \sqrt{\bar{\alpha}_t}x_0, (1-\bar{\alpha}_t)\mathbf{I})$, where $\mathcal{N}(x_t; \mu, \mathbf{I})$ denotes a Gaussian distribution over $x$ with mean $\mu$ and covariance $\mathbf{I}$. Here, $\bar{\alpha}_t = \prod_{i=1}^{t} \alpha_i$, and the sequence $\{\alpha_t\}_{t=1}^{N}$ is a predefined hyperparameter schedule that controls the noise level at each step. The model, $\epsilon_\theta(x_t, t)$, is trained to predict the noise $\epsilon$ added at step $t$ by optimizing a simplified mean-squared error objective:

$$\mathcal{L}(\theta) = \mathbb{E}_{t,x_0,\epsilon}[||\epsilon - \epsilon_\theta(\sqrt{\bar{\alpha}_t}x_0 + \sqrt{1-\bar{\alpha}_t}\epsilon, t)||^2]. \tag{2}$$

### 2.2.1 GUIDED DIFFUSION

Guided diffusion techniques steer the generation process towards desired conditions $y$.

**Classifier Guidance** Dhariwal & Nichol (2021) uses the gradient of a separately trained classifier $p_\phi(y|x_t)$, to adjust the noise prediction $\epsilon_\theta$ at each sampling step:

$$\hat{\epsilon}_\theta(x_t, t, y) = \epsilon_\theta(x_t, t) + s \cdot \Sigma_t \nabla_{\boldsymbol{x}_t} \log p_\phi(y|x_t), \tag{3}$$

where $s$ is the guidance scale that controls the strength of the conditioning.

**Classifier-Free Guidance** Ho & Salimans (2022) avoids a separate classifier by training a single conditional model, $\epsilon_\theta(x_t, t, y))$, that is jointly trained on the condition y and a null token $\emptyset$. Guidance is then achieved by interpolating between the conditional and unconditional predictions:

$$\hat{\epsilon}_\theta(x_t, t, y) = w\epsilon_\theta(x_t, t, y) + (1-w)\epsilon_\theta(x_t, t, \emptyset), \tag{4}$$

where $w$ is the guidance weight.

## 3 METHODS

We approach safe offline RL by framing it as a conditional generative modeling task over trajectories. Our aim is to learn a generative model $p_\theta(\tau|R(\tau), C(\tau))$. This model captures the distribution of trajectories conditioned on rewards and costs. We train the model parameters $\theta$ by maximizing the conditional log-likelihood on the dataset $\mathcal{D}$:

$$\max_\theta \mathbb{E}_{\tau \sim \mathcal{D}} \left[ \log p_\theta(\tau \mid R(\tau), C(\tau)) \right]. \tag{5}$$

By learning this conditional distribution, we establish a foundation for the ultimate goal: leveraging the model $p_\theta$ during inference to generate trajectories that achieve high rewards while satisfying the safety constraint $C(\tau) \leq l$. This approach to setting a predefined time-varying cost limit is practical because our model generates a new trajectory from noise at each task timestep. Therefore, the specific cost limit applied for the current generation does not predetermine the limits for subsequent generations. Subsequently, we denote $R(\tau), C(\tau)$ as $R, C$.

### 3.1 SAFE DUAL-GUIDE DIFFUSER

We propose the *Safe Dual-Guide Diffuser (SDGD)*, a novel framework that generates safe, high-reward trajectories by decoupling the guidance mechanisms for safety and performance. This design is motivated by the distinct nature of the two objectives. Reward optimization is a maximization problem ("more is better"), making it well-suited for gradient-based classifier guidance (CG). In contrast, safety is a hard constraint satisfaction problem ("stay below the limit"), which is more reliably handled by directly conditioning the generative process via classifier-free guidance (CFG).

Our approach begins by training a conditional diffusion model to serve as a cost-conditioned generative base. This model, denoted $p_\theta(\tau_{t-1}|\tau_t, C)$, is trained using CFG to generate trajectories that inherently adhere to a specific cumulative cost target $C$. To incorporate reward optimization, we then introduce a separately trained reward classifier $p_\phi(R|\tau_t)$, which estimates the cumulative reward $R$ given the noisy trajectory $\tau_t$. We combine these two components using Bayes' rule to form a joint distribution conditioned on both cost and reward:

$$p(\tau_{t-1}|\tau_t, R, C) \propto p(\tau_{t-1}|\tau_t, C)p(R|\tau_{t-1}, C), \tag{6}$$

The first term is the cost-conditioned denoising step from our base CFG model, while the second term introduces the reward guidance from the classifier. Following standard practice, we assume the reward prediction is primarily dependent on the trajectory itself, approximating $p(R|\tau_{t-1}, C) \approx p(R|\tau_{t-1})$. We treat Eq. (6) as a perturbed Gaussian distribution. Intuitively, the cost-conditioned diffusion model provides the base Gaussian, while the reward classifier contributes a small gradient-based shift of the mean. Following the classifier-guidance formulation of Dhariwal & Nichol (2021), we approximate the reward gradient $\nabla_{\tau_{t-1}} \log p_\phi(R \mid \tau_{t-1})$ using the classifier. This yields an approximate conditional distribution $p(\tau_{t-1} \mid \tau_t, R, C)$ of the form:

$$\begin{aligned}
\log(p(\tau_{t-1}|\tau_t, R, C)) &= \log[p(\tau_{t-1}|\tau_t, C)p(R|\tau_{t-1})] \\
&\approx \log p(z) + Z, z \sim \mathcal{N}(\mu(\tau_t, C) + \Sigma g, \Sigma).
\end{aligned} \tag{7}$$

where $Z$ is a normalizing constant and $g = \nabla_{\tau_{t-1}} \log p_\phi(r|\tau_{t-1})|_{\tau_{t-1}=\mu(\tau_t, C)}$. Thus the base mean $\mu(\tau_t, C)$ (parameterized by $\epsilon_\theta(\tau_t, t, C)$) is shifted by the reward gradient $g$, producing a cost-conditioned and reward-guided denoising step. *For derivations, please refer to Appendix A.1.*

For the trajectory representation itself, we employ a Sparse-State Dense-Action (SS-DA) Chen et al. (2024) representation. This structure is designed to balance the long-horizon planning capabilities characteristic of hierarchical methods with the computational efficiency of a single, non-hierarchical diffusion model. We quantitatively validate our choice in our ablation studies (Appendix, Table 8).

### 3.2 FEASIBLE TRAJECTORY RE-LABELING

While gradients of the reward classifier $p_\phi$ can guide a diffusion model towards higher rewards, training $p_\phi$ directly on raw cumulative rewards from a dataset $\mathcal{D}$ poses a significant safety challenge. The issue arises because the offline data may contain trajectories that achieve high rewards at the expense of incurring high costs. A naively trained classifier can therefore learn a spurious correlation between features indicative of high rewards and those associated with unsafe, high-cost behaviors. Consequently, using the classifier for reward guidance can inadvertently steer the generation process into these dangerous regions. To mitigate this risk and decouple reward-seeking from unsafe behavior, we introduce a Feasible Trajectory Re-labeling (FTR) mechanism.

Our FTR approach begins by relabeling the cumulative reward for any trajectory with an initially unsafe segment. We formalize this by defining feasibility based strictly on the costs incurred within the first $f$ steps (where $f$ is a predefined hyperparameter, *Feasible Length*) as follows:

**Infeasible if:** $\exists t \in \{0, \ldots, f-1\}, c_t > 0$        **Feasible if:** $\forall t \in \{0, \ldots, f-1\}, c_t = 0$    (8)

The rationale behind this definition is to use the reward signal to strongly penalize trajectories that begin unsafely, while relying on the dedicated cost-based guidance (CFG) to manage potential costs later in trajectories.

To determine a sufficiently large penalty $r_{us}$, our strategy is to establish worst-case reward bounds for both feasible and infeasible trajectories. This ensures that after re-labeling, even the highest-reward infeasible trajectory is valued less than the lowest-reward feasible one. We define these bounds by

considering extreme scenarios over a trajectory of length $L$. We set an upper bound $R^*$ on the reward of any infeasible trajectory by assuming it attains the maximum per-step reward $r_{\max}$ at every step. Symmetrically, we define a lower bound $R^o$ for feasible trajectories by assuming they receive the minimum per-step reward $r_{\min}$ at every step. These bounds are calculated as follows:

$$\textbf{Infeasible Upper Bound: } R^* = \frac{r_{\max}(1 - \gamma^{L-1})}{1 - \gamma}, \quad \textbf{Feasible Lower Bound: } R^o = \frac{r_{\min}(1 - \gamma^{L-1})}{1 - \gamma}, \quad (9)$$

To ensure that the penalized reward of even the highest-reward infeasible trajectory remains strictly lower than that of the lowest-reward feasible one, we impose the condition $R^* + r_{\text{us}} < R^o$. This yields the requirement for the safety penalty $r_{\text{us}} < \frac{(r_{\min} - r_{\max})(1 - \gamma^{L-1})}{1 - \gamma}$.

In practice, we observe $r_{\min}$ and $r_{\max}$ from $\mathcal{D}$ and compute a value for $r_{\text{us}}$ satisfying this bound. The modified reward $\tilde{R}(\tau)$, used for training the reward classifier $p_\phi$ is then defined as:

$$\tilde{R}(\tau) = R(\tau) + r_{\text{us}} \cdot \mathbb{I}(\tau \text{ is Infeasible}). \tag{10}$$

By training $p_\phi$ to predict these modified rewards $\tilde{R}(\tau)$, the classifier learns to associate trajectory segments that lead to initial infeasibility with significantly lower reward. Consequently, the gradient $\nabla_{\tau_t} \log p_\phi(r|\tau_t)$ is strongly discouraged from steering towards trajectories that begin unsafely, mitigating the risk of improving cost.

### 3.3 THEORETICAL GUARANTEES

We establish that SDGD achieves formal safety and near-optimality guarantees. Our theoretical analysis relies on a standard concentration assumption on the offline dataset (see Assumption 1 in the Appendix). This assumption bounds the estimation error between the real environment $T$ and the cost function $c$, and the empirical model $(\hat{T}, \hat{c})$ learned from the datasets. We denote the expected cumulative costs under a trajectory distribution $q$ in these respective environments as $J_{c,T}(q) = \mathbb{E}_{\tau \sim q, T}[\sum_{t=0}^{L} \gamma^t c(s_t, a_t)]$ and $J_{\hat{c},\hat{T}}(q) = \mathbb{E}_{\tau \sim q, \hat{T}}[\sum_{t=0}^{L} \gamma^t \hat{c}(s_t, a_t)]$.

The bound on their difference depends on several key quantities, including the maximum per-step costs ($c_{\max}$ and $\hat{c}_{\max}$) under the real and empirical functions. It is also governed by the policy divergence, $\delta_\pi = \sup_s D_{TV}(\pi(\cdot|s)||\pi_\beta(\cdot|s))$, which measures the maximum deviation from the behavior policy $\pi_\beta$, and the counts $N(s, a)$ for each state-action pair in datasets.

**Proposition 1** (Cost Performance Bound). *Let $q$ be the trajectory distribution generated by SDGD. The gap between the true and empirical expected cost is bounded as follows:*

$$|J_{c,T}(q) - J_{\hat{c},\hat{T}}(q)| \leq \frac{4c_{\max}\gamma}{(1-\gamma)^2}\delta_\pi + \mathbb{E}_{\pi_\beta,T}\left[\sum_{t=0}^{L} \frac{\gamma^t C_{c,\delta}}{\sqrt{N(s_t, a_t)}}\right] + \frac{2\hat{c}_{\max}}{1-\gamma}\mathbb{E}_{\pi_\beta,\hat{T}}\left[\sum_{t=0}^{L} \frac{\gamma^t C_{T,\delta}}{\sqrt{N(s_t, a_t)}}\right],$$

$$(11)$$

*where $C_{T,\delta}$ and $C_{c,\delta}$ are constants depending on the concentration properties of $T(s'|s, a)$ and $c(s, a)$, respectively, and $\delta \in (0, 1)$. Proof is in Appendix A.2.*

According to Proposition 1, the estimation error of the expected cost is mainly controlled by two types of terms. The first, policy divergence error (proportional to $\delta_\pi$), arises from the distribution shift from the behavior policy. Controlling this error requires keeping the learned policy close to the dataset, which presents a trade-off between ensuring safety and discovering more optimal out-of-distribution policies. The second, finite-sample error (the terms with $\frac{1}{\sqrt{N(s_t, a_t)}}$), reflects model uncertainty due to limited data coverage. Crucially, it reveals our safety mechanism: minimizing policy divergence anchors the total error to a lower bound set by the behavior policy's uncertainty.

Beyond providing safety, we also prove that SDGD's performance is competitive with that of the true optimal safe policy, $\pi_{\text{safe}}^*$. This establishes a formal guarantee of near-optimality. To formalize this, we denote the expected cumulative reward under a trajectory distribution $q^*$ as $J_{r,T}(q^*) = \mathbb{E}_{\tau \sim q^*, T}[\sum_{t=0}^{L} \gamma^t r(s_t, a_t)]$.

**Theorem 1** (Optimality Guarantee). *Let $q^*$ be the trajectory distribution generated by SDGD. The expected reward in the real environment, $J_{r,T}(q^*)$, is lower-bounded as follows:*

$$J_{r,T}(q^*) \geq J_{r,T}(\pi_{\text{safe}}^*) - \left(b_r(q^*) + b_r(\pi_{\text{safe}}^*)\right), \tag{12}$$

*where $b_r(\cdot)$ is the reward "reality-gap" bound, which, analogous to the cost bound, depends on policy divergence and finite-sample errors (formally defined in Proposition 3 in the appendix).*

Theorem 1 provides a crucial insight by decomposing the sub-optimality gap into two distinct components encapsulated by the $b_r(\cdot)$ terms. The first component, $b_r(q^*)$, represents the estimation error of our learned policy, arising from its divergence from the dataset and finite-sample errors, mirroring the factors in the safety bound. More subtly, the second component, $b_r(\pi^*_{\text{safe}})$, quantifies the estimation error for the true optimal safe policy. This term reveals a fundamental limitation of offline learning: it will be large if optimal safe behaviors are poorly represented in the dataset, regardless of how well our algorithm learns. This decomposition is critical because it shows that SDGD's performance is not only determined by the quality of the learned policy but is also fundamentally constrained by the quality and coverage of the provided data. An offline algorithm can only be as good as the best policy supported by its dataset.

### 3.4 PRACTICAL IMPLEMENTATION

To obtain the optimal trajectory distribution, we adopt *Diffuser* Janner et al. (2022) and *Decision Diffuser* Ajay et al. (2022), two milestone trajectory generation studies that use CG and CFG to maximize rewards, respectively. A useful perspective, articulated in Janner et al. (2022), frames planning as guided sampling. The practical objective in Eq. (6) can be seen as sampling from a modified distribution biased towards desirable trajectories:

$$\tilde{p}_\theta(\tau) \propto p_\theta(\tau)h(\tau), \tag{13}$$

where the function $h(\tau)$ is a guidance function encoding optimality preferences.

For Safe RL, an ideal distribution would favor trajectories that achieve high rewards while strictly satisfying safety constraints. Let $\mathcal{O}_t$ be a binary random variable denoting the optimality of timestep $t$ of a trajectory, with $p(\mathcal{O}_{1:T} = 1 \mid \tau) = \prod_{t=1}^{T} \exp(r(s_t, a_t))\mathbb{I}(C_t \leq l)$. We can express the distribution over optimal trajectories by setting $h(\tau) = p(\mathcal{O}_{1:T} = 1 \mid \tau)$ in Eq. (13):

$$\tilde{p}_\theta(\tau) = p_\theta(\tau \mid \mathcal{O}_{1:T} = 1) \propto p_\theta(\tau)p(\mathcal{O}_{1:T} = 1 \mid \tau) \propto p_\theta(\tau)\prod_{t=1}^{T} \exp(r(s_t, a_t))\mathbb{I}(C_t \leq l). \tag{14}$$

Here, $\mathbb{I}$ is an indicator function that equals 1 if the cumulative cost $C_t$ up to timestep $t$ is less than or equal to a cost limit $l$, and 0 otherwise. The pseudocode for the guided planning method is given in Appendix B, Algorithm 1, 2. *Please see Appendix B for more hyperparameter details.*

## 4 EXPERIMENTS

In this section, we evaluate the performance of the SDGD on the DSRL benchmark Liu et al. (2023a). This benchmark encompasses 38 diverse tasks across the simulation environments: Safety-Gymnasium Ji et al. (2023), Bullet-Safety-Gym Gronauer (2022), and MetaDrive Li et al. (2022). Our goal is to empirically answer the following questions: 1) Can SDGD effectively learn policies from offline data that achieve high rewards while reliably satisfying safety constraints? 2) Can SDGD achieve adaptation to different cost limits? 3) Can SDGD adapt to cost limits that vary dynamically within a single evaluation episode? 4) What is each SDGD core component's contribution to performance and safety? Empirically, SDGD achieves a decision efficiency of 2.1 Hz on an NVIDIA A6000 GPU, significantly faster than prior methods like TREBI (1.1 Hz) and HD (1.0 Hz).

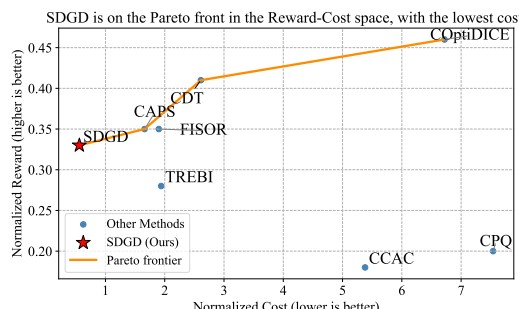

Figure 4: Aggregate reward-cost performance, averaged over all 38 DSRL tasks. Safety is defined as a normalized cost $\leq 1$. SDGD is the only method to achieve an average cost within this safe region, establishing a superior trade-off.

**Evaluation Setups.** We evaluate performance using normalized reward $R_{\text{normalized}}$ and normalized cost $C_{\text{normalized}}$ adhering to the normalization procedures from the DSRL benchmark Liu et al. (2023a).

Table 1: Normalized DSRL Liu et al. (2023a) benchmark results. ↑ means the higher the better. ↓ means the lower the better. Each value is averaged over 20 evaluation episodes and 3 seeds. Gray: Unsafe agents. **Bold**: Safe agents whose normalized cost is smaller than 1. **Blue**: Safe agents with the highest reward.

| Task | COptiDICE | | CPQ | | CDT | | TREBI | | FISOR | | CAPS | | CCAC | | SDGD (ours) | |
|---|---|---|---|---|---|---|---|---|---|---|---|---|---|---|---|---|
| | reward↑ | cost↓ | reward↑ | cost↓ | reward↑ | cost↓ | reward↑ | cost↓ | reward↑ | cost↓ | reward↑ | cost↓ | reward↑ | cost↓ | reward↑ | cost↓ |
| CarButton1 | -0.16 | 4.63 | 0.22 | 40.06 | 0.17 | 7.05 | 0.07 | 3.75 | **-0.19** | **0.85** | **-0.02** | **0.73** | 0.47 | 33.25 | **-0.03** | **0.63** |
| CarButton2 | -0.17 | 3.40 | 0.08 | 19.03 | 0.23 | 12.87 | **-0.03** | **0.97** | **0.00** | **0.25** | **-0.09** | **0.60** | 0.49 | 35.48 | **-0.05** | **0.52** |
| CarCircle1 | 0.70 | 17.69 | 0.22 | 17.40 | 0.48 | 6.91 | **0.14** | **0.00** | 0.68 | 11.48 | 0.51 | 5.97 | -0.46 | 4.00 | **0.14** | **0.76** |
| CarCircle2 | 0.78 | 26.56 | **0.55** | **0.36** | 0.56 | 11.92 | **0.22** | **0.00** | 0.58 | 8.32 | 0.50 | 3.83 | -0.22 | 20.09 | **0.34** | **0.15** |
| CarGoal1 | 0.43 | 2.81 | 0.33 | 4.93 | 0.60 | 3.15 | 0.41 | 1.16 | 0.47 | 1.31 | 0.33 | 1.51 | 0.84 | 5.62 | **0.31** | **0.63** |
| CarGoal2 | 0.19 | 2.83 | 0.10 | 6.31 | 0.45 | 6.05 | 0.13 | 1.16 | **0.04** | **0.87** | 0.10 | 2.14 | 0.94 | 16.97 | **0.10** | **0.58** |
| CarPush1 | 0.21 | 1.28 | **0.08** | **0.77** | 0.27 | 2.12 | 0.26 | 1.03 | 0.27 | 1.75 | **0.16** | **0.51** | 0.36 | 6.58 | **0.21** | **0.92** |
| CarPush2 | 0.10 | 4.55 | -0.03 | 10.00 | 0.16 | 4.60 | 0.12 | 2.65 | **0.20** | **0.96** | 0.10 | 1.89 | -0.07 | 19.62 | **0.04** | **0.70** |
| PointButton1 | 0.09 | 3.34 | 0.46 | 11.88 | 0.06 | 1.53 | 0.07 | 3.23 | 0.06 | 1.24 | **-0.04** | **0.60** | -0.48 | 3.21 | **0.04** | **0.70** |
| PointButton2 | 0.10 | 2.77 | 0.52 | 15.04 | 0.25 | 7.18 | **0.08** | **0.60** | 0.12 | 1.13 | 0.13 | 4.96 | -0.47 | 1.96 | **0.05** | **0.92** |
| PointCircle1 | 0.85 | 18.08 | -0.26 | 3.20 | **0.53** | **0.42** | 0.41 | 0.00 | 0.27 | 15.95 | 0.19 | 1.74 | -0.52 | 10.13 | 0.29 | 1.49 |
| PointCircle2 | 0.86 | 28.66 | 0.10 | 11.50 | 0.44 | 0.90 | 0.40 | 6.62 | 0.72 | 14.96 | 0.38 | 0.00 | 0.50 | 2.68 | **0.49** | **0.56** |
| PointGoal1 | 0.50 | 5.17 | 0.51 | 0.20 | 0.36 | 1.10 | 0.38 | 2.21 | 0.67 | 3.67 | 0.20 | 0.78 | 0.77 | 5.21 | **0.43** | **0.86** |
| PointGoal2 | 0.44 | 7.31 | 0.53 | 10.57 | 0.29 | 1.77 | 0.30 | 2.58 | **0.18** | **0.56** | 0.30 | 1.40 | -0.98 | 1.52 | **0.21** | **0.89** |
| PointPush1 | 0.15 | 2.98 | 0.22 | 3.94 | **0.16** | **0.48** | 0.21 | 1.16 | **0.27** | **0.72** | 0.14 | 1.08 | **0.06** | **0.46** | 0.11 | 0.76 |
| PointPush2 | -0.04 | 4.88 | 0.09 | 5.32 | 0.13 | 2.54 | 0.11 | 3.7 | 0.27 | 1.9 | 0.11 | 2.09 | -0.08 | **0.88** | **0.05** | **0.81** |
| AntVel | 1.00 | 10.29 | -1.01 | 0.00 | 0.89 | 3.08 | 0.31 | 0.00 | 0.89 | 0.00 | 0.89 | 0.38 | 0.32 | 0.00 | **0.90** | **0.01** |
| HalfCheetahVel | **0.64** | **0.00** | 0.08 | 2.56 | 0.56 | 1.90 | **0.87** | **0.23** | **0.89** | **0.00** | 0.88 | 0.31 | 0.85 | 0.92 | 0.83 | 0.00 |
| HopperVel | 0.10 | 3.77 | 0.16 | 13.40 | **0.17** | **0.86** | 0.08 | 7.39 | **0.17** | **0.70** | 0.17 | 0.01 | 0.11 | 0.68 | **0.63** | **0.34** |
| SwimmerVel | 0.58 | 23.64 | 0.31 | 11.58 | 0.67 | 1.47 | 0.42 | 1.31 | **-0.02** | **0.01** | 0.43 | 7.01 | 0.00 | 2.44 | **0.47** | **0.15** |
| Walker2dVel | 0.13 | 2.63 | 0.23 | 0.36 | **0.74** | **0.32** | **0.10** | **0.72** | 0.39 | 1.10 | **0.80** | **0.01** | -0.01 | 0.00 | 0.10 | 0.79 |
| AntRun | 0.57 | 1.05 | **0.02** | **0.00** | 0.72 | 1.60 | 0.69 | 2.54 | **0.36** | **0.39** | 0.52 | 1.09 | **0.14** | **0.00** | **0.50** | **0.04** |
| BallRun | 0.62 | 6.30 | 0.24 | 1.90 | **0.20** | **0.00** | 0.28 | 1.71 | **0.18** | **0.08** | 0.13 | 0.00 | 0.04 | 0.80 | **0.26** | **0.14** |
| CarRun | **0.87** | **0.00** | 0.98 | 4.32 | 0.34 | 1.44 | 0.96 | 1.90 | **0.76** | **0.00** | **0.97** | **0.42** | 1.68 | 18.13 | **0.97** | **0.03** |
| DroneRun | 0.65 | 7.80 | 0.28 | 3.02 | **0.58** | **0.32** | 0.40 | 4.99 | **0.17** | **0.00** | 0.35 | 13.57 | 0.71 | 10.94 | **0.35** | **0.58** |
| AntCircle | 0.20 | 7.63 | **0.00** | **0.00** | 0.45 | 4.06 | 0.64 | 6.17 | **0.20** | **0.00** | 0.31 | 0.00 | **0.52** | **0.05** | 0.20 | 0.98 |
| BallCircle | 0.70 | 4.34 | 0.62 | 1.08 | **0.38** | **0.67** | 0.67 | 1.29 | **0.29** | **0.00** | 0.58 | 0.30 | **0.73** | **0.15** | 0.49 | 0.81 |
| CarCircle | 0.46 | 3.53 | 0.71 | 0.00 | **0.72** | **0.38** | 0.64 | 2.11 | **0.39** | **0.00** | 0.62 | 0.42 | **0.72** | **0.23** | 0.40 | 0.90 |
| DroneCircle | 0.30 | 2.23 | -0.22 | 1.83 | 0.51 | 1.57 | 0.56 | 4.49 | **0.49** | **0.09** | 0.45 | 0.18 | 0.31 | 1.45 | 0.09 | 2.56 |
| easysparse | 0.89 | 9.94 | **-0.06** | **0.10** | **-0.03** | **0.00** | 0.26 | 2.26 | **0.41** | **0.08** | 0.00 | 0.49 | -0.06 | 0.16 | **0.62** | **0.02** |
| easymean | 0.57 | 6.20 | **-0.06** | **0.10** | **0.38** | **0.00** | 0.00 | 0.41 | **0.42** | **0.21** | 0.01 | 0.78 | -0.06 | 0.00 | **0.60** | **0.39** |
| easydense | 0.47 | 4.70 | **-0.06** | **0.10** | 0.50 | 0.57 | 0.00 | 0.28 | **0.48** | **0.86** | 0.10 | 0.49 | -0.06 | 0.10 | **0.71** | **0.82** |
| mediumsparse | 0.88 | 4.97 | **-0.06** | **0.10** | **0.40** | **0.11** | 0.10 | 0.94 | **0.39** | **0.09** | **0.41** | **0.23** | -0.08 | 0.16 | 0.17 | 0.10 |
| mediummean | 0.87 | 5.00 | **-0.06** | **0.10** | 0.63 | 2.53 | 0.19 | 1.33 | **0.59** | **0.85** | 0.85 | 2.85 | -0.06 | 0.00 | 0.19 | 0.11 |
| mediumdense | 0.89 | 4.68 | **-0.06** | **0.25** | 0.93 | 4.08 | 0.08 | 0.74 | **0.50** | **0.05** | 0.56 | 1.01 | -0.07 | 0.16 | 0.20 | 0.11 |
| hardsparse | 0.38 | 3.56 | **-0.05** | **0.10** | 0.31 | 0.98 | 0.11 | 1.11 | 0.32 | 0.14 | 0.57 | 1.81 | -0.05 | 0.10 | **0.44** | **0.32** |
| hardmean | 0.33 | 3.59 | **-0.05** | **0.16** | 0.09 | 0.01 | -0.05 | 0.09 | 0.25 | 0.36 | 0.17 | 0.16 | -0.05 | 0.10 | **0.45** | **0.06** |
| harddense | 0.23 | 2.50 | **-0.04** | **0.10** | 0.23 | 0.03 | 0.02 | 0.14 | 0.30 | 0.47 | 0.36 | 1.85 | -0.04 | 0.10 | **0.44** | **0.35** |
| **Safe / Optimal** | 2 | 0 | 14 | 1 | 16 | 4 | 15 | 2 | 27 | 7 | 21 | 4 | 20 | 3 | 36 | 21 |
| **Average** | 0.46 | 6.72 | 0.20 | 7.54 | 0.41 | 2.61 | 0.28 | 1.94 | 0.35 | 1.90 | 0.35 | 1.66 | 0.18 | 5.38 | **0.33** | **0.56** |

Safety is strictly defined as achieving $C_{\text{normalized}} \leq 1$. For a rigorous and unified safety assessment, we evaluate all methods under a strict, uniform absolute cost limit of 10 across all tasks. This represents a more challenging evaluation compared to the original DSRL benchmark's protocol, which uses varying and often higher thresholds for different environments (e.g., {10, 20, 40} for BulletGym and MetaDrive, and {20, 40, 80} for SafetyGym). Our evaluation protocol follows that of DSRL.

**Baselines.** We compare our method with the following six baselines: 1) *COptiDICE* Lee et al. (2022): A stationary **DI**stribution **C**orrection **E**stimation (DICE) based safe offline RL method that builds on *OptiDICE* Lee et al. (2021). 2) *CPQ* Xu et al. (2022): A penalized based method that considers out-of-distribution action. 3) *CDT* Liu et al. (2023b): A Decision Transformer based method that considers safety constraints. 4) *TREBI* Lin et al. (2023): A diffusion-based method that utilizes classifier-guidance sampling to sample safe trajectories. 5) *FISOR* Zheng et al. (2024): A feasibility-guided method that utilizes a diffusion model to estimate policy. 6) *CAPS* Chemingui et al. (2025): A framework that adapts to cost limits via a set of policies and dynamically switching between their actions to satisfy the constraint. 7) *CCAC* Guo et al. (2025a): A actor-critic method that learns a single adaptive policy by taking the cost limit as a direct input to the actor and critics.

**Main Results.** Across 38 OSRL tasks, the results in Table 1 demonstrate that SDGD establishes a new state-of-the-art, significantly outperforming all baselines in both safety and performance. SDGD achieves the highest safety compliance, successfully completing 36 out of 38 tasks while satisfying the strict normalized cost constraint. Furthermore, among the safe agents, SDGD obtains

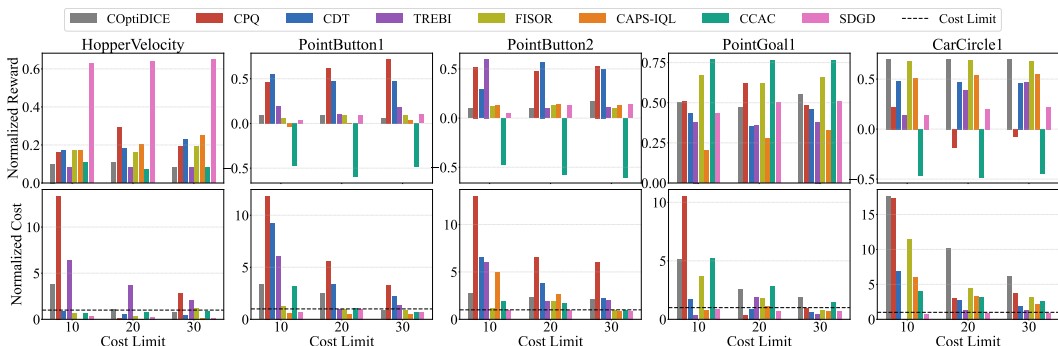

Figure 5: Performance comparison across different cost limits (10, 20, 30) in five representative tasks. The top and bottom rows show the normalized reward and cost, respectively.

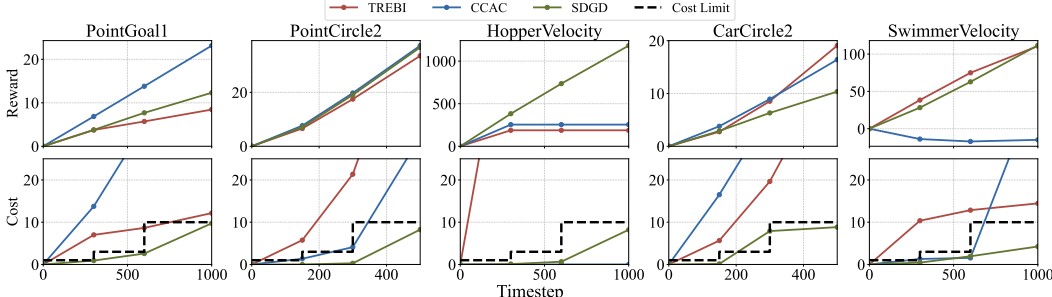

Figure 6: Performance under dynamically time-varying cost limits. The experiment evaluates the adaptability of TREBI, CCAC, and SDGD by varying the cost limit within a single episode (from 1, to 3, to 10). The dashed line represents the changing cost limit.

state-of-the-art reward performance on 21 tasks, which is three times more than any competitor. The actor-critic baselines designed for adaptability, CAPS and CCAC, demonstrate moderate safety, satisfying constraints in 21 and 20 tasks, respectively. The other generative models, CDT, TREBI, and FISOR, achieve safety in more tasks than older approaches but still fail in challenging scenarios where SDGD succeeds. Conversely, earlier methods like CPQ are often overly conservative, yielding suboptimal rewards, while COptiDICE is frequently unsafe. As shown in Figure 4, SDGD lies on the aggregate Pareto frontier, establishing a superior trade-off by dominating the safe region.

### 4.1 ABLATION STUDY AND ANALYSIS

**Ablation on Cost Limits.** To assess our method's adaptability to varying safety requirements, we evaluated SDGD against baselines under several cost limits (Figure 5). The results confirm SDGD's superior ability to balance performance and safety. While most baselines become either unsafe or overly conservative, SDGD consistently secures high rewards while satisfying the specified constraints. Crucially, this adaptability is achieved with a single model; SDGD, along with CDT, TREBI, CAPS, and CCAC, handles all cost limits with one set of weights.

**Time-Varying Limits.** A key challenge in real-world scenarios is adapting to dynamically changing safety constraints. We designed a demanding time-varying evaluation where the cost limit changes within a single episode (see Figure 6 for setup.) SDGD is the only method that successfully adapts to these fluctuations, maintaining safety while opportunistically maximizing rewards. This highlights its unique robustness for dynamic environments, a critical failure point for the compared baselines.

**Dual-Guide Diffuser.** We performed an ablation study to validate the effectiveness of our proposed dual-guide mechanism 7. The results confirm that both components are essential and complementary. Removing either the cost-controlling (CFG) or the reward-optimizing (CG) guidance leads to a failure in satisfying both objectives simultaneously, underscoring the necessity of our dual-guide design for achieving safe, high-performing policies. *For detailed results please see Appendix C, Table 6.*

**Hyperparameter Choices.** To validate our dual-guide design, we performed an ablation study on the reward guidance scale and our trajectory re-labeling mechanism, controlled by Feasible Length (f).

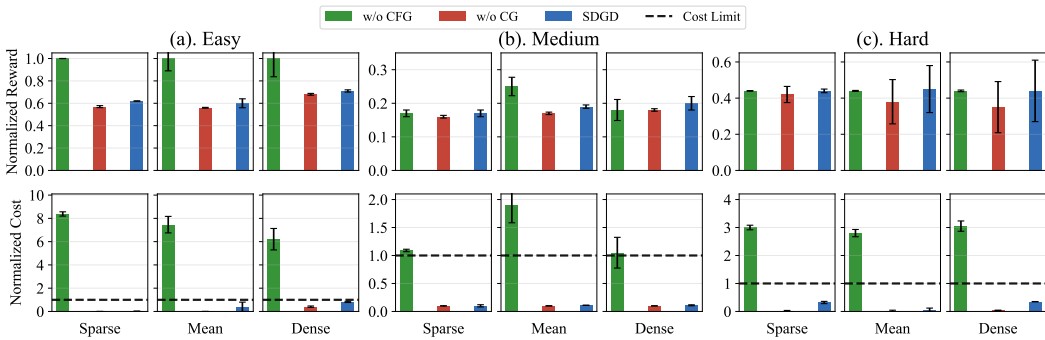

Figure 7: Ablation study on the dual-guide components. The SDGD is compared against two variants: one without Classifier Guidance (w/o CG) and another without Classifier-Free Guidance (w/o CFG).

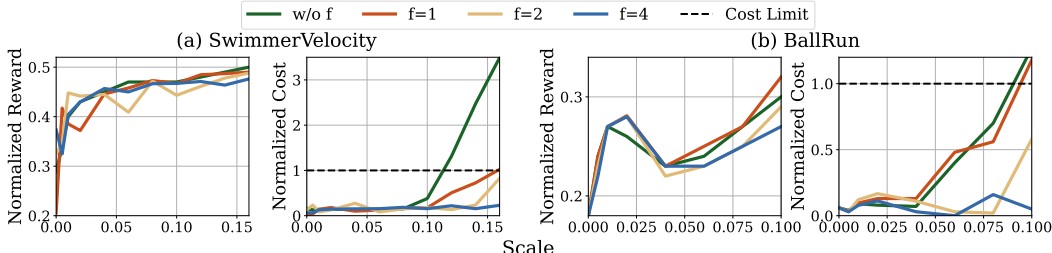

Figure 8: Ablation on the reward guidance scale and Feasible Length $f$ for trajectory re-labeling. Without re-labeling ('w/o $f$'), stronger guidance boosts rewards but violates the cost limit, whereas re-labeling ($f>0$) maintains safety, with larger $f$ values encouraging conservative behavior.

The results in Figure 8 confirm re-labeling is essential for preventing safety violations under strong reward guidance, and that larger $f$ values encourage more conservative policies.

## 5 RELATED WORK

**Safe RL.** Online approaches predominantly employ Lagrangian methods Chow et al. (2018); Ding et al. (2020) integrated with trust region optimization. Constrained policy optimization methods, including CPO Achiam et al. (2017), FOCOPS Zhang et al. (2020), and PCPO Yang et al. (2020), extend TRPO Schulman (2015) with theoretical safety guarantees through constraints. In offline Safe RL, foundational work addressed the challenge of satisfying a fixed safety constraint while combating distribution shift, employing diverse strategies such as Lagrangian methods (BCQ-Lag) Stooke et al. (2020), distribution correction (COptiDICE) Lee et al. (2022), penalty-based approaches (CPQ) Xu et al. (2022), and feasibility-guided methods (FISOR) Zheng et al. (2024). SDQC Yang et al. (2025) focuses on representation learning to decouple features relevant to safety from those for task performance. The subsequent challenge evolved to adapting to varying cost limits without retraining. This was tackled by two primary paradigms: actor-critic frameworks, like the policy-conditioning CCAC Guo et al. (2025a), the policy-switching CAPS Chemingui et al. (2025), and constraint-conditioned implicit Q-learning (C2IQL) LIU et al. (2025); and generative models, like the target-conditioned transformer CDT Liu et al. (2023b) and diffusion-based TREBI Lin et al. (2023).

**Trajectory Optimization with Planning.** Model-based trajectory optimization Nagabandi et al. (2020); Hamrick et al. (2020) has evolved into two main paradigms: autoregressive Transformers Chen et al. (2021); Huang et al. (2024) and diffusion models, with the latter generating entire trajectories simultaneously through planners like Diffuser Janner et al. (2022); Ajay et al. (2022), Diffusion Policy Chi et al. (2023), and latent-space variants Li (2023); Yan et al. (2024).

## 6 CONCLUSION AND FUTURE WORK

We presented the Safe Dual-Guide Diffuser (SDGD), a novel framework that decouples safety and reward optimization to achieve state-of-the-art performance on the DSRL benchmark. Future work will focus on improving trajectory generation efficiency to enable real-world deployment.

## ETHICS STATEMENT

This research aims to advance the field of safe reinforcement learning, with the primary goal of developing an algorithm that can handle effectively while adhering to critical safety constraints. The proposed method, SDGD, is designed to enhance the safety and reliability of autonomous systems, which we believe is a positive societal contribution, particularly for safety-critical applications like autonomous driving and robotics.

Our work is purely computational and utilizes the offline reinforcement learning paradigm. This approach inherently mitigates ethical concerns associated with online learning in the real world, as it avoids potentially hazardous environmental interactions during the training phase. The experiments are conducted on publicly available and well-established benchmarks (DSRL), which consist of simulated environments and do not involve human subjects, animals, or personally identifiable information.

## REPRODUCIBILITY STATEMENT

We are committed to ensuring the reproducibility of our research. All experiments were conducted using the publicly available DSRL benchmark, which includes datasets from Safety-Gymnasium Bullet-Safety-Gym, and MetaDrive. Our evaluation protocol, including normalization and safety criteria, is detailed in Section 4. The core methodology of our approach, SDGD, is described in Section 3. Key implementation details, including pseudocode for the training and sampling procedures (Algorithms 1 and 2) and a comprehensive list of hyperparameters (Table 2), are provided in Appendix B. Our implementation builds upon the publicly available codebases for Diffuser and Decision Diffuser, as cited in Appendix B.1.

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

## A APPENDIX

We first analyze the feasibility of using two kinds of guidance to constrain the cost of trajectories and improve the reward.

### A.1 PROOF OF DUAL-GUIDE DIFFUSION PROCESS

This section details the derivation for guiding a diffusion model's reverse process using both a reward condition $R$ and a cost condition $C$. We aim to find the form of $p(\tau_{t-1}|\tau_t, R, C)$ based on the cost-conditional reverse process $p(\tau_{t-1}|\tau_t, C)$ and a reward classifier $p(R|\tau_t)$.

By the Markov property, a key observation is: $p(R|\tau_{t-1}, \tau_t, C) = p(R|\tau_{t-1}, C)$.

*Derivation as follows:*

$$
\begin{aligned}
p(R|\tau_{t-1}, \tau_t, C) &= \frac{p(R, \tau_t|\tau_{t-1}, C)}{p(\tau_t|\tau_{t-1}, C)} \\
&= \frac{p(\tau_t|\tau_{t-1}, C) \cdot (R|\tau_{t-1}, C)}{p(\tau_t|\tau_{t-1}, C)} \\
&= p(R|\tau_{t-1}, C)
\end{aligned}
\tag{15}
$$

Assumption: The reward $R$ is conditionally independent of $\tau_t$ given $\tau_{t-1}$ and $C$.

Substituting this back into Equation 15:

$$
p(\tau_{t-1}|\tau_t, R, C) = \frac{p(\tau_{t-1}, \tau_t, R, C)}{p(\tau_t, R, C)} \tag{16}
$$

$$
= \frac{p(R|\tau_{t-1}, \tau_t, C)p(\tau_{t-1}, \tau_t, C)}{p(\tau_t, C)p(R|\tau_t, C)} \tag{17}
$$

$$
= \frac{p(R|\tau_{t-1}, \tau_t, C)p(\tau_{t-1}|\tau_t, C)\cancel{p(\tau_t, C)}}{\cancel{p(\tau_t, C)}p(R|\tau_t, C)} \tag{18}
$$

$$
= \frac{p(R|\tau_{t-1}, \tau_t, C)p(\tau_{t-1}|\tau_t, C)}{p(R|\tau_t, C)} \tag{19}
$$

$$
= \frac{p(R|\tau_{t-1}, C)p(\tau_{t-1}|\tau_t, C)}{p(R|\tau_t, C)} \tag{20}
$$

The term $p(R|\tau_t, C)$ is constant with respect to $\tau_{t-1}$. Thus, Equation 20 becomes:

$$
p(\tau_{t-1}|\tau_t, R, C) \propto p(\tau_{t-1}|\tau_t, C)p(R|\tau_{t-1}, C) \tag{21}
$$

We assume $p(R|\tau_{t-1}, C) = p(R|\tau_{t-1})$, meaning the reward prediction from the noisy trajectory $\tau_t$ is independent of the cost condition $C$. This simplifies the gradient calculation. We approximate $\log p(R|\tau_{t-1}, C)$ using a first-order Taylor expansion around $\tau_{t-1} = \mu$:

$$\log p(R|\tau_{t-1}, C) \approx \log p(R|\tau_t, C) + (\tau_{t-1} - \mu)\nabla_{\tau_{t-1}} \log p(R|\tau_{t-1}, C)\big|_{\tau_{t-1}=\mu}$$

$$= \log p(R|\tau_t) + (\tau_{t-1} - \mu)g, \quad g = \nabla_{\tau_t} \log p(R|\tau_{t-1})|_{\tau_{t-1}=\mu} \tag{22}$$

For the parameterized reverse process, let $p_\theta(\tau_{t-1}|\tau_t, C) = \mathcal{N}(\mu_\theta(\tau_t, C), \Sigma(t))$ with denoising network $\epsilon_\theta(\tau_t, C)$. Using a reward network $p_\phi(R|\tau_t)$, we derive:

$$\log p(\tau_{t-1}|\tau_t, R, C) = \log[Z p_\theta(\tau_{t-1}|\tau_t, C)p_\phi(r|\tau_{t-1}, C)]$$

$$\approx -\frac{1}{2}(\tau_{t-1} - \mu_\theta(\tau_t, C))^\top \Sigma^{-1}(\tau_{t-1} - \mu_\theta(\tau_t, C)) + (\tau_{t-1} - \tau_t)g + Z_1$$

$$\approx -\frac{1}{2}(\tau_{t-1} - \mu_\theta(\tau_t, C))^\top \Sigma^{-1}(\tau_{t-1} - \mu_\theta(\tau_t, C)) + (\tau_{t-1} - \mu_\theta(\tau_t, C))g + Z_2$$

$$= -\frac{1}{2}(\tau_{t-1} - \mu_\theta(\tau_t, C) - \Sigma g)^\top \Sigma^{-1}(\tau_{t-1} - \mu_\theta(\tau_t, C) - \Sigma g) + Z_3 \tag{23}$$

$$= \log p(z) + C_4, z \sim \mathcal{N}(\mu_\theta(x_t, C) + \Sigma g, \Sigma) \tag{24}$$

Thus, the guided reverse process becomes:

$$p(\tau_{t-1}|\tau_t, R, C) = \mathcal{N}(\mu_\theta(\tau_t, C) + \Sigma g, \ \Sigma) \tag{25}$$

## A.2 Theoretical Analysis

This section provides the detailed theoretical analysis and proofs for the guarantees presented in Section 3.3.

### A.2.1 Assumptions and Supporting Lemmas

Our analysis relies on the following standard assumption regarding the concentration of the empirical model learned from the offline dataset.

**Assumption 1** (Concentration of Empirical Model). *For all state-action pairs $(s, a)$ in the offline dataset, we assume that with probability at least $1 - \delta$, the empirical estimates $\hat{T}$, $\hat{c}$, and $\hat{r}$ satisfy:*

$$\|T(\cdot|s, a) - \hat{T}(\cdot|s, a)\|_1 \leq \frac{C_{T,\delta}}{\sqrt{N(s, a)}}$$

$$|c(s, a) - \hat{c}(s, a)| \leq \frac{C_{c,\delta}}{\sqrt{N(s, a)}}$$

$$|r(s, a) - \hat{r}(s, a)| \leq \frac{C_{r,\delta}}{\sqrt{N(s, a)}}$$

*where $N(s, a)$ is the count of the pair $(s, a)$ in the dataset, and $C_{T,\delta}, C_{c,\delta}, C_{r,\delta}$ are constants.*

We also establish two supporting lemmas that will be used in our main proofs.

**Lemma 1** (MDP TVD Bound). *Consider two MDPs with the same transition dynamics $T$ but different policies $\pi_1, \pi_2$, inducing state-action distributions $p_1^t(s, a)$ and $p_2^t(s, a)$. Let the total variation distance (TVD) of their initial state-action distributions be $D_{\text{TV}}(p_1^0\|p_2^0) = \delta_0$, and assume the TVD between the policies is uniformly bounded by $\sup_s D_{\text{TV}}(\pi_1(\cdot|s)\|\pi_2(\cdot|s)) \leq \delta_\pi$. Then, the TVD at any timestep $t$ is bounded by:*

$$D_{\text{TV}}(p_1^t(s, a)\|p_2^t(s, a)) \leq \delta_0 + t\delta_\pi$$

*Proof.* We prove this by induction. The base case for $t = 0$ holds by definition. For the inductive step, we bound the TVD at timestep $t$ based on the TVD at $t - 1$.

$$D_{\mathrm{TV}}(p_1^t \| p_2^t) = \frac{1}{2} \sum_{s,a} \left| p_1^t(s,a) - p_2^t(s,a) \right|$$

$$= \frac{1}{2} \sum_{s,a} \left| \sum_{s',a'} T(s|s',a') \left( p_1^{t-1}(s',a')\pi_1(a|s) - p_2^{t-1}(s',a')\pi_2(a|s) \right) \right|$$

$$\leq \frac{1}{2} \sum_{s,a} \sum_{s',a'} T(s|s',a') \left| p_1^{t-1}(s',a')\pi_1(a|s) - p_2^{t-1}(s',a')\pi_2(a|s) \right|$$

$$= \frac{1}{2} \sum_{s,a} \sum_{s',a'} T(s|s',a') \left| (p_1^{t-1} - p_2^{t-1})\pi_1 + p_2^{t-1}(\pi_1 - \pi_2) \right|$$

$$\leq \frac{1}{2} \sum_{s',a'} |p_1^{t-1}(s',a') - p_2^{t-1}(s',a')| \sum_{s,a} T(s|s',a')\pi_1(a|s)$$

$$+ \frac{1}{2} \sum_{s',a'} p_2^{t-1}(s',a') \sum_{s,a} T(s|s',a')|\pi_1(a|s) - \pi_2(a|s)|$$

$$= D_{\mathrm{TV}}(p_1^{t-1} \| p_2^{t-1}) + \mathbb{E}_{(s',a') \sim p_2^{t-1}} \left[ D_{\mathrm{TV}}(\pi_1(\cdot|s) \| \pi_2(\cdot|s)) \right]$$

$$\leq D_{\mathrm{TV}}(p_1^{t-1} \| p_2^{t-1}) + \delta_\pi$$

Applying this recurrence relation from $t$ down to 0, we arrive at $\delta_t \leq \delta_0 + t\delta_\pi$. $\qquad\square$

**Lemma 2** (Branched Costs Bound). *For any two trajectory distributions $p_1$ and $p_2$, the difference in their expected cumulative discounted costs is bounded by:*

$$|J_{c,T}(p_1) - J_{c,T}(p_2)| \leq 2c_{\max} \left( \frac{\delta_0}{1-\gamma} + \frac{\delta_\pi \gamma}{(1-\gamma)^2} \right)$$

*Proof.* Inspired by the branched returns bound from Janner et al. (2019), we can write the difference in expected costs as:

$$|J_{c,T}(p_1) - J_{c,T}(p_2)| = \left| \sum_t \sum_{s,a} \gamma^t \left( p_1^t(s,a) - p_2^t(s,a) \right) c(s,a) \right|$$

$$\leq \sum_t \gamma^t \sum_{s,a} |p_1^t(s,a) - p_2^t(s,a)| \cdot |c(s,a)|$$

$$\leq c_{\max} \sum_t \gamma^t \left( 2D_{\mathrm{TV}}(p_1^t \| p_2^t) \right)$$

Applying Lemma 1, we get:

$$|J_{c,T}(p_1) - J_{c,T}(p_2)| \leq 2c_{\max} \sum_t \gamma^t (\delta_0 + t\delta_\pi)$$

$$= 2c_{\max} \left( \delta_0 \sum_t \gamma^t + \delta_\pi \sum_t t\gamma^t \right)$$

$$= 2c_{\max} \left( \frac{\delta_0}{1-\gamma} + \frac{\delta_\pi \gamma}{(1-\gamma)^2} \right)$$

$\qquad\square$

### A.2.2    PROOF OF COST PERFORMANCE BOUND

This section provides the full proof for Proposition 1 from the main text.

**Proposition 2** (Cost Performance Bound). *Let $q$ be the trajectory distribution generated by SDGD, and $\pi_\beta$ be the behavior policy. The gap between the true and empirical expected cost is bounded by:*

$$|J_{c,T}(q) - J_{\hat{c},\hat{T}}(q)| \le \frac{4c_{\max}\gamma}{(1-\gamma)^2}\delta_\pi + \mathbb{E}_{\pi_\beta,T}\left[\sum_{t=0}^{L}\frac{\gamma^t C_{c,\delta}}{\sqrt{N(s_t,a_t)}}\right] + \frac{2\hat{c}_{\max}}{1-\gamma}\mathbb{E}_{\pi_\beta,\hat{T}}\left[\sum_{t=0}^{L}\frac{\gamma^t C_{T,\delta}}{\sqrt{N(s_t,a_t)}}\right]$$

*where $\delta_\pi = \sup_s D_{\mathrm{TV}}(\pi(\cdot|s)\|\pi_\beta(\cdot|s))$ is the maximum policy divergence.*

*Proof.* Our strategy is to use the triangle inequality, introducing the performance of the behavior policy $\pi_\beta$ under both the true $(T,c)$ and empirical $(\hat{T},\hat{c})$ models as an intermediate bridge. This decomposes the total error into three terms:

$$|J_{c,T}(q) - J_{\hat{c},\hat{T}}(q)| \le \underbrace{|J_{c,T}(q) - J_{c,T}(\pi_\beta)|}_{\text{Term A}} + \underbrace{|J_{c,T}(\pi_\beta) - J_{\hat{c},\hat{T}}(\pi_\beta)|}_{\text{Term B}} + \underbrace{|J_{\hat{c},\hat{T}}(\pi_\beta) - J_{\hat{c},\hat{T}}(q)|}_{\text{Term C}}$$

We now bound each term individually.

**Bounding Terms A and C (Policy Divergence):** These terms measure the performance gap between our policy $q$ and the behavior policy $\pi_\beta$ under the same model. Since the initial state distribution is the same ($\delta_0 = 0$), we can apply Lemma 2 to get:

$$|J(q) - J(\pi_\beta)| \le \frac{2c_{\max}\gamma}{(1-\gamma)^2}\delta_\pi$$

This bound applies to both Term A (under model $T, c$) and Term C (under model $\hat{T}, \hat{c}$, using $\hat{c}_{\max}$). We use the more general $c_{\max}$ for both, so the sum of bounds for A and C is $\frac{4c_{\max}\gamma}{(1-\gamma)^2}\delta_\pi$.

**Bounding Term B (Reality Gap):** This term captures the reality gap for the behavior policy due to finite data. We decompose it further to isolate errors from the cost function and the dynamics:

$$|J_{c,T}(\pi_\beta) - J_{\hat{c},\hat{T}}(\pi_\beta)| \le \underbrace{|J_{c,T}(\pi_\beta) - J_{\hat{c},T}(\pi_\beta)|}_{\text{Cost Function Error}} + \underbrace{|J_{\hat{c},T}(\pi_\beta) - J_{\hat{c},\hat{T}}(\pi_\beta)|}_{\text{Dynamics Error}}$$

Using Assumption 1, we bound each part:

**Cost Function Error**:

$$|J_{c,T}(\pi_\beta) - J_{\hat{c},T}(\pi_\beta)| \le \mathbb{E}_{\tau\sim\pi_\beta,T}\left[\sum_{t=0}^{L}\gamma^t|c(s_t,a_t) - \hat{c}(s_t,a_t)|\right] \le \mathbb{E}_{\tau\sim\pi_\beta,T}\left[\sum_{t=0}^{L}\frac{\gamma^t C_{c,\delta}}{\sqrt{N(s_t,a_t)}}\right]$$

**Dynamics Error**:

$$|J_{\hat{c},T}(\pi_\beta) - J_{\hat{c},\hat{T}}(\pi_\beta)| = \left|\mathbb{E}_{\pi_\beta,T}\left[\sum_{t=0}^{L}\gamma^t\hat{c}(s_t,a_t)\right] - \mathbb{E}_{\pi_\beta,\hat{T}}\left[\sum_{t=0}^{L}\gamma^t\hat{c}(s_t,a_t)\right]\right|$$

$$\le \frac{2\hat{c}_{\max}}{1-\gamma}\mathbb{E}_{\pi_\beta,\hat{T}}\left[\sum_{t=0}^{L}\frac{\gamma^t C_{T,\delta}}{\sqrt{N(s_t,a_t)}}\right]$$

Combining the bounds for Terms A, B, and C completes the proof. $\square$

### A.3 THEORETICAL GUARANTEE FOR OPTIMALITY

This section provides the proof for the near-optimality guarantee of SDGD (Theorem 1). We first establish a performance bound for reward, analogous to the one for cost.

**Proposition 3** (Reward Performance Bound). *Let $q$ be the trajectory distribution generated by SDGD. The gap between the true and empirical expected reward is bounded by:*

$$|J_{r,T}(q) - J_{\hat{r},\hat{T}}(q)| \le \frac{4r_{\max}\gamma}{(1-\gamma)^2}\delta_\pi + \mathbb{E}_{\pi_\beta,T}\left[\sum_{t=0}^{L}\frac{\gamma^t C_{r,\delta}}{\sqrt{N(s_t,a_t)}}\right] + \frac{2\hat{r}_{\max}}{1-\gamma}\mathbb{E}_{\pi_\beta,T}\left[\sum_{t=0}^{L}\frac{\gamma^t C_{T,\delta}}{\sqrt{N(s_t,a_t)}}\right]$$

*The proof is analogous to that of Proposition 1, replacing the cost function $c$ with the reward function $r$. For brevity, we denote this entire error bound as $b_r(q)$.*

This proposition establishes that the true expected reward is bounded by the empirical reward: $J_{r,T}(q) \geq J_{\hat{r},\hat{T}}(q) - b_r(q)$. We now state and prove the main theorem.

**Theorem 2** (Optimality Guarantee). *Let $q^*$ be the trajectory distribution generated by SDGD, and let $\pi^*_{safe}$ be the optimal safe policy in the true environment. The true expected reward of the SDGD policy is lower-bounded by:*

$$J_{r,T}(q^*) \geq J_{r,T}(\pi^*_{safe}) - \big(b_r(q^*) + b_r(\pi^*_{safe})\big)$$

*Proof.* Our objective is to bound the sub-optimality gap, $J_{r,T}(\pi^*_{safe}) - J_{r,T}(q^*)$. We again use the triangle inequality to decompose this gap by introducing the performance of both policies on the empirical model $(\hat{r}, \hat{T})$:

$$J_{r,T}(\pi^*_{safe}) - J_{r,T}(q^*) = \underbrace{\Big(J_{r,T}(\pi^*_{safe}) - J_{\hat{r},\hat{T}}(\pi^*_{safe})\Big)}_{\text{Term 1: Reality Gap for } \pi^*_{safe}}$$

$$+ \underbrace{\Big(J_{\hat{r},\hat{T}}(\pi^*_{safe}) - J_{\hat{r},\hat{T}}(q^*)\Big)}_{\text{Term 2: Empirical Performance Gap}}$$

$$+ \underbrace{\Big(J_{\hat{r},\hat{T}}(q^*) - J_{r,T}(q^*)\Big)}_{\text{Term 3: Reality Gap for } q^*}$$

We can bound each term as follows:

1. **Terms 1 & 3 (Reality Gaps):** Using Proposition 3, we can bound the reality gaps for both policies:

$$J_{r,T}(\pi^*_{safe}) - J_{\hat{r},\hat{T}}(\pi^*_{safe}) \leq |J_{r,T}(\pi^*_{safe}) - J_{\hat{r},\hat{T}}(\pi^*_{safe})| \leq b_r(\pi^*_{safe})$$
$$J_{\hat{r},\hat{T}}(q^*) - J_{r,T}(q^*) \leq |J_{\hat{r},\hat{T}}(q^*) - J_{r,T}(q^*)| \leq b_r(q^*)$$

2. **Term 2 (Empirical Performance Gap):** This term compares the two policies on the empirical model. By design, the SDGD algorithm is optimized to find a high-performing safe trajectory distribution $q^*$ on this model. Therefore, we expect its performance to be at least as good as any other safe policy on the same model, i.e., $J_{\hat{r},\hat{T}}(q^*) \geq J_{\hat{r},\hat{T}}(\pi^*_{safe})$. This implies that Term 2 is non-positive: $J_{\hat{r},\hat{T}}(\pi^*_{safe}) - J_{\hat{r},\hat{T}}(q^*) \leq 0$.

Combining these bounds, we have:

$$J_{r,T}(\pi^*_{safe}) - J_{r,T}(q^*) \leq b_r(\pi^*_{safe}) + b_r(q^*)$$

Rearranging this inequality gives the lower bound on the performance of SDGD, concluding the proof. $\square$

**Interpretation.** Theorem 1 formally states that the performance of SDGD is competitive with the true optimal safe policy. The sub-optimality gap is controlled by two main factors: the quality and coverage of the dataset (which determines the size of the reality gap bounds, $b_r$) and the divergence between the learned policies and the behavior policy. The $b_r(q^*)$ term depends on the divergence of our learned policy from the behavior data, $\delta_\pi(q^*, \pi_\beta)$, while the $b_r(\pi^*_{safe})$ term depends on the divergence of the true optimal safe policy from the behavior data, $\delta_\pi(\pi^*_{safe}, \pi_\beta)$, which reflects how well the optimal safe behaviors are represented in the dataset. This provides a strong theoretical justification that SDGD is designed to achieve not only safety but also near-optimal performance, constrained by the available data.

# B IMPLEMENTATION DETAILS

This section outlines the implementation details including the pseudocode of our method and the hyperparameters.

| **Algorithm 1** Training | **Algorithm 2** Sampling |
|---|---|
| 1: **Given:** Diffusion model $(\mu_\theta(\tau_t), \Sigma_t)$, dataset $\mathcal{D}$, Reward label $R$, cost label $C$. | 1: **Given:** Current state s, gradient scale $s$, classifier-free guidance strength $w$. |
| 2: // Feasible trajectory re-labeling | 2: $\tau_T \sim \mathcal{N}(\mathbf{0}, \mathbf{I})$ |
| 3: Retrieve the dataset $\mathcal{D}$ to get $r_{\max}$ and $r_{\min}$. | 3: **for** $t = T, \ldots, 1$ **do** |
| 4: Calculate $r_{\mathrm{us}}$ and relabel $R(\tau_0)$. | 4: $\quad \nabla_{\tau_t} R(\tau_t) \leftarrow p_\phi(\tau_t)$ |
| 5: **repeat** | 5: $\quad \hat{\epsilon}_t = (1 + w)\epsilon_\theta(\tau_t, C) - w\epsilon_\theta(\tau)$ |
| 6: $\quad (\tau_0, R(\tau_0), C(\tau_0)) \sim \mathcal{D}$ | 6: $\quad (\mu_\theta(\tau_{t-1}), \Sigma_\theta(\tau_{t-1})) \leftarrow Denoise(\tau_t, \hat{\epsilon}_t)$ |
| 7: $\quad \epsilon \sim \mathcal{N}(\mathbf{0}, \mathbf{I})$ | 7: $\quad \tau_{t-1} \sim \mathcal{N}(\mu_\theta(\tau_{t-1}) + s\nabla_{\tau_t} R(\tau_t), \Sigma_\theta(\tau_{t-1}))$ |
| 8: $\quad t \sim Uniform(\{1, .., T\})$. | 8: $\quad$ // Replace the first state of the $\tau_{t-1}$ |
| 9: $\quad \tau_t = \sqrt{\bar{\alpha}_t}\tau_0 + (1 - \bar{\alpha}_t)\epsilon$ | 9: $\quad \tau_{t-1} \leftarrow$ s |
| 10: $\quad$ Update $\mu_\theta$ using Eq. (2). | 10: **end for** |
| 11: **until** converged | 11: Execute first action of trajectory: $a \leftarrow \tau_0$. |

### B.1 PRACTICAL IMPLEMENTATION

We represent the noise model $\epsilon_\theta$ with a temporal U-Net follow Diffuser Janner et al. (2022) [1] and Decision Diffuser Ajay et al. (2022) [2]. We use a guidance free scale $w \in \{1, 2, 4, 6, 8\}$ and a guidance scale $s \in \{0.005, 0.01, 0.02, 0.04\}$ but the exact choice varies by task.

Table 2: Hyperparameters setting.

| Hyperparameter | Value |
|---|---|
| Optimizer | Adam |
| Learning Rate | 2e-4 |
| Batch Size | 32 |
| Training Steps (noise model) | 1e6 |
| Training Steps (classifier) | 2e5 |
| Horizon | 64 |
| Feasible Length | 2 |
| Diffusion Steps | 20 |
| Model Layers (noise model) | 128 * (1,4,8) |
| Model Layers (classifier) | 32 * (1,4,8) |

## C EXPERIMENTS DETAILS

This section provides supplementary benchmark results, ablation studies, and implementation details to support and validate the main findings in our paper and ensure reproducibility.

### C.1 COMPREHENSIVE BENCHMARK RESULTS ACROSS DIFFERENT COST LIMITS

To supplement the main paper's results (Table 1), which analyze performance under a strict cost limit ($L = 10$), this section provides the complete, per-task benchmark results for all methods under relaxed cost limits of $L = 20$ (Table 3) and $L = 30$ (Table 4).

These tables provide empirical support for the robustness of our method, SDGD. They demonstrate that its superior safety-performance trade-off is maintained across different safety requirements, and its effectiveness is not confined to a single constraint level. The evaluation includes both adaptive and non-adaptive baselines.

### C.2 RESULTS ON MORE BASELINES

In this section, we present a more detailed comparison against additional fundamental offline RL baselines: BC and BCQ-Lag Fujimoto et al. (2019). The per-environment results are detailed in

---
[1] https://github.com/jannerm/diffuser

[2] https://github.com/anuragajay/decision-diffuser/tree/main

Table 3: Normalized DSRL Liu et al. (2023a) benchmark results under cost limit $L = 20$. ↑ means the higher the better. ↓ means the lower the better. Each value is averaged over 20 evaluation episodes and 3 seeds. Gray: Unsafe agents. **Bold**: Safe agents whose normalized cost is smaller than 1. **Blue**: Safe agents with the highest reward.

| Task | COptiDICE | | CPQ | | CDT | | TREBI | | FISOR | | CAPS | | CCAC | | SDGD (ours) | |
|---|---|---|---|---|---|---|---|---|---|---|---|---|---|---|---|---|
| | reward↑ | cost↓ | reward↑ | cost↓ | reward↑ | cost↓ | reward↑ | cost↓ | reward↑ | cost↓ | reward↑ | cost↓ | reward↑ | cost↓ | reward↑ | cost↓ |
| CarButton1 | -0.02 | 1.32 | 0.42 | 16.56 | 0.05 | 3.20 | 0.09 | 1.09 | **0.00** | **0.27** | -0.03 | 0.18 | 0.42 | 18.45 | **0.00** | **0.68** |
| CarButton2 | -0.13 | 3.11 | 0.26 | 23.05 | -0.09 | 2.57 | 0.06 | 1.43 | **-0.02** | **0.20** | -0.06 | 1.19 | 0.45 | 20.88 | **0.00** | **0.89** |
| CarCircle1 | 0.70 | 10.18 | -0.18 | 3.01 | 0.47 | 2.74 | 0.39 | 1.38 | 0.75 | 6.25 | 0.54 | 3.30 | 0.48 | 7.30 | **0.20** | **0.95** |
| CarCircle2 | 0.80 | 13.88 | **0.52** | **0.21** | 0.59 | 5.00 | 0.59 | 1.58 | 0.62 | 5.34 | 0.50 | 2.15 | 0.56 | 2.50 | **0.36** | **0.50** |
| CarGoal1 | 0.29 | 1.15 | 0.80 | 1.62 | **0.42** | **0.97** | **0.44** | **0.62** | 0.42 | 0.97 | 0.25 | 0.73 | 0.86 | 3.30 | **0.32** | **0.67** |
| CarGoal2 | **0.17** | **0.27** | 0.44 | 5.35 | 0.16 | 1.38 | 0.25 | 1.05 | **0.03** | **0.14** | 0.09 | 1.29 | 0.89 | 10.82 | **0.10** | **0.29** |
| CarPush1 | 0.23 | 1.51 | 0.10 | 1.25 | **0.15** | **0.80** | 0.18 | 0.66 | **0.24** | **0.08** | 0.13 | 0.09 | 0.47 | 1.11 | **0.19** | **0.30** |
| CarPush2 | 0.13 | 1.54 | 0.21 | 7.18 | 0.14 | 2.49 | **0.11** | **0.19** | 0.00 | 0.30 | 0.04 | 1.52 | 0.17 | 8.83 | **0.04** | **0.53** |
| PointButton1 | 0.09 | 2.55 | 0.62 | 5.62 | 0.47 | 3.41 | **0.09** | **1.00** | **0.09** | **0.58** | 0.00 | 0.50 | 0.73 | 5.98 | **0.09** | **1.00** |
| PointButton2 | 0.10 | 2.31 | 0.47 | 6.51 | 0.57 | 3.79 | 0.10 | 1.90 | 0.10 | 2.02 | 0.14 | 2.70 | 0.62 | 7.64 | **0.13** | **0.99** |
| PointCircle1 | 0.86 | 9.20 | -0.26 | 1.60 | **0.51** | **0.11** | 0.23 | 1.43 | 0.63 | 7.84 | 0.41 | 1.02 | 0.62 | 4.13 | **0.43** | **0.89** |
| PointCircle2 | 0.86 | 14.34 | **0.44** | **0.43** | 0.48 | 0.63 | 0.47 | 0.93 | 0.43 | 10.56 | 0.43 | 0.34 | 0.52 | 1.74 | **0.49** | **0.78** |
| PointGoal1 | 0.47 | 2.55 | **0.62** | **0.35** | 0.35 | 0.86 | 0.36 | 1.93 | 0.65 | 2.34 | 0.28 | 1.09 | 0.79 | 3.19 | **0.50** | **0.72** |
| PointGoal2 | 0.40 | 2.71 | 0.21 | 1.64 | 0.46 | 1.99 | 0.37 | 2.00 | 0.33 | 1.04 | **0.25** | **0.78** | 0.81 | 7.90 | **0.23** | **0.72** |
| PointPush1 | 0.16 | 1.79 | **0.35** | **0.33** | 0.11 | 1.32 | **0.33** | **0.49** | 0.28 | 0.28 | 0.09 | 0.42 | 0.16 | 1.27 | **0.19** | **0.95** |
| PointPush2 | 0.07 | 1.49 | **0.06** | **0.49** | 0.18 | 0.95 | **0.21** | **0.27** | 0.29 | 1.53 | 0.12 | 0.84 | 0.08 | 1.22 | **0.05** | **0.41** |
| AntVel | 1.00 | 6.62 | **-1.01** | **0.00** | 0.92 | 3.14 | 0.26 | 0.03 | 0.88 | 0.00 | **0.92** | **0.60** | -1.01 | 0.00 | 0.91 | 0.01 |
| HalfCheetahVel | 0.68 | 0.00 | 0.30 | 0.46 | 0.61 | 2.71 | **0.90** | **0.96** | 0.88 | 0.00 | **0.90** | **0.68** | 0.83 | 1.00 | 0.83 | 0.00 |
| HopperVel | **0.09** | **0.80** | 0.28 | 0.01 | 0.18 | 0.58 | 0.08 | 3.74 | 0.13 | 0.65 | 0.22 | 0.13 | 0.07 | 0.75 | **0.64** | **0.22** |
| SwimmerVel | 0.66 | 14.90 | **0.04** | **0.90** | **0.67** | **0.79** | 0.50 | 1.14 | 0.05 | 0.51 | 0.50 | 4.95 | -0.08 | 0.01 | 0.49 | 0.79 |
| Walker2dVel | 0.12 | 1.13 | **0.02** | **0.11** | 0.42 | 2.25 | **0.09** | **0.18** | 0.50 | 0.83 | **0.80** | **0.32** | -0.01 | 0.00 | 0.10 | 0.54 |
| AntRun | **0.58** | **0.39** | 0.03 | 0.00 | 0.69 | 1.10 | **0.65** | **0.79** | 0.43 | 0.11 | 0.62 | 0.84 | 0.02 | 0.00 | 0.50 | 0.00 |
| BallRun | 0.68 | 3.38 | 0.29 | 0.65 | 0.28 | 0.52 | 0.42 | 1.24 | 0.22 | 0.18 | 0.13 | 1.46 | **0.33** | **0.90** | 0.30 | 0.75 |
| CarRun | **0.90** | **0.00** | 0.97 | 4.60 | 0.55 | 0.58 | 0.96 | 1.07 | 0.75 | 0.07 | 0.97 | 0.24 | 1.76 | 9.08 | **0.98** | **0.40** |
| DroneRun | 0.65 | 3.86 | 0.12 | 0.01 | **0.60** | **0.93** | 0.70 | 3.01 | 0.32 | 0.80 | 0.30 | 7.04 | 0.32 | 2.69 | 0.35 | 0.29 |
| AntCircle | 0.18 | 4.70 | **0.22** | **0.00** | 0.46 | 1.76 | 0.41 | 2.76 | 0.23 | 0.00 | 0.40 | 0.18 | **0.54** | **0.66** | 0.24 | 0.90 |
| BallCircle | 0.72 | 2.26 | **0.64** | **0.14** | 0.47 | 0.31 | 0.72 | 0.86 | 0.35 | 0.00 | 0.71 | 0.66 | **0.78** | **0.58** | 0.66 | 0.97 |
| CarCircle | 0.47 | 1.98 | **0.71** | **0.00** | **0.72** | **0.62** | 0.64 | 0.66 | 0.47 | 0.01 | 0.68 | 0.71 | 0.71 | 0.85 | 0.37 | 0.96 |
| DroneCircle | 0.26 | 1.17 | -0.18 | 1.14 | **0.53** | **0.68** | 0.51 | 2.74 | 0.47 | 0.01 | **0.53** | **0.69** | 0.27 | 0.93 | 0.13 | 1.56 |
| easysparse | 0.90 | 4.51 | **-0.06** | **0.00** | 0.04 | 0.05 | 0.06 | 0.53 | 0.32 | 0.08 | 0.01 | 0.28 | -0.06 | 0.08 | **0.67** | **0.18** |
| easymean | 0.57 | 3.04 | **-0.07** | **0.05** | 0.52 | 0.08 | -0.01 | 0.18 | 0.44 | 0.10 | 0.01 | 0.38 | -0.06 | 0.00 | **0.73** | **0.76** |
| easydense | 0.59 | 2.88 | **-0.06** | **0.05** | 0.61 | 0.71 | 0.00 | 0.14 | 0.57 | 0.65 | 0.10 | 0.25 | -0.06 | 0.05 | **0.74** | **0.56** |
| mediumsparse | 0.33 | 1.05 | **-0.08** | **0.08** | 0.90 | 1.45 | -0.08 | 0.05 | 0.37 | 0.02 | **0.49** | **0.05** | -0.08 | 0.08 | 0.30 | 0.15 |
| mediummean | 0.87 | 2.40 | **-0.08** | **0.05** | 0.74 | 1.72 | 0.19 | 0.66 | **0.62** | **0.05** | 0.85 | 1.47 | -0.06 | 0.00 | 0.25 | 0.14 |
| mediumdense | 0.86 | 2.28 | **-0.08** | **0.05** | 0.85 | 2.03 | 0.09 | 0.37 | 0.52 | 0.06 | **0.80** | **0.78** | -0.07 | 0.08 | 0.20 | 0.06 |
| hardsparse | 0.41 | 2.33 | **-0.04** | **0.08** | 0.25 | 0.78 | 0.11 | 0.56 | **0.63** | **0.31** | 0.54 | 0.89 | -0.05 | 0.05 | 0.53 | 0.45 |
| hardmean | 0.46 | 2.35 | **-0.05** | **0.05** | 0.11 | 0.28 | -0.04 | 0.04 | 0.26 | 0.00 | 0.23 | 0.07 | -0.05 | 0.05 | **0.45** | **0.03** |
| harddense | **0.18** | **0.98** | **-0.04** | **0.08** | 0.43 | 1.53 | -0.02 | 0.07 | 0.34 | 0.34 | 0.51 | 1.64 | -0.04 | 0.05 | **0.44** | **0.18** |
| **Safe/Optimal** | 6 | 1 | 25 | 3 | 19 | 5 | 22 | 6 | 30 | 3 | 25 | 8 | 19 | 3 | 37 | 13 |
| **Average** | 0.46 | 3.50 | 0.18 | 2.20 | 0.44 | 1.50 | 0.30 | 1.05 | 0.39 | 1.17 | 0.38 | 1.12 | 0.33 | 3.27 | **0.37** | **0.56** |

Table 5. The results show that our method, SDGD, significantly outperforms these baselines in safely maximizing rewards.

Our ablation studies on the DSRL benchmark Liu et al. (2023a) provide granular insights into the framework components. Table 6 presents the comparative analysis of two architectural variants: 1) *w/o CG* removes classifier guidance while retaining safety-centric trajectory generation capabilities, and 2) *w/o CFG* eliminates classifier-free guidance but maintains high reward optimization capacity.

## C.3 ABLATION

### C.3.1 SWAPPING GUIDANCE ROLES

To prove our core hypothesis that Classifier Guidance (CG) is unreliable for hard safety constraints and Classifier-Free Guidance (CFG) is the necessary solution, we conducted a critical "swap" ablation study. This new baseline (Algorithm 3)), termed "CG-Cost / CFG-Reward", precisely inverts our proposed SDGD design.

The results of this experiment, shown in Table 7. The variant is unsafe on every representative task, validating our claim that this mechanism is unsuitable for safety constraints.

Table 4: Normalized DSRL Liu et al. (2023a) benchmark results under cost limit $L = 30$. ↑ means the higher the better. ↓ means the lower the better. Each value is averaged over 20 evaluation episodes and 3 seeds. Gray: Unsafe agents. **Bold**: Safe agents whose normalized cost is smaller than 1. **Blue**: Safe agents with the highest reward.

| Task | COptiDICE reward↑ | cost↓ | CPQ reward↑ | cost↓ | CDT reward↑ | cost↓ | TREBI reward↑ | cost↓ | FISOR reward↑ | cost↓ | CAPS reward↑ | cost↓ | CCAC reward↑ | cost↓ | SDGD (ours) reward↑ | cost↓ |
|---|---|---|---|---|---|---|---|---|---|---|---|---|---|---|---|---|
| CarButton1 | -0.01 | 1.20 | 0.36 | 12.17 | 0.17 | 2.53 | **0.16** | **0.59** | -0.07 | 0.32 | 0.00 | 0.07 | 0.53 | 10.22 | **0.00** | **0.53** |
| CarButton2 | -0.07 | 1.83 | 0.35 | 14.15 | 0.05 | 2.46 | 0.11 | 1.12 | -0.03 | 0.32 | -0.08 | 0.80 | 0.46 | 11.25 | **0.00** | **0.59** |
| CarCircle1 | 0.70 | 6.12 | -0.07 | 3.81 | 0.46 | 1.90 | 0.47 | 1.35 | 0.71 | 3.20 | 0.55 | 2.20 | 0.56 | 5.09 | **0.22** | **0.93** |
| CarCircle2 | 0.77 | 9.47 | 0.06 | 11.58 | 0.62 | 3.66 | **0.36** | **0.50** | 0.66 | 5.28 | 0.51 | 1.46 | 0.55 | 2.90 | **0.37** | **0.62** |
| CarGoal1 | **0.49** | **0.60** | 0.84 | 1.75 | 0.53 | 1.33 | **0.54** | **0.27** | 0.37 | 0.53 | 0.30 | 0.45 | 0.84 | 2.11 | **0.32** | **0.45** |
| CarGoal2 | 0.20 | 1.01 | 0.61 | 3.56 | **0.22** | **0.21** | 0.39 | 1.08 | **0.03** | **0.02** | 0.09 | 0.29 | 0.91 | 6.03 | **0.10** | **0.20** |
| CarPush1 | **0.23** | **0.28** | 0.18 | 0.59 | 0.12 | 0.51 | 0.24 | 0.19 | 0.20 | 0.21 | 0.20 | 0.18 | 0.40 | 1.01 | **0.30** | **0.89** |
| CarPush2 | 0.10 | 1.23 | 0.13 | 5.80 | 0.19 | 1.42 | **0.12** | **0.49** | **0.13** | **0.23** | 0.06 | 0.99 | 0.15 | 5.12 | **0.04** | **0.18** |
| PointButton1 | **0.06** | **0.85** | 0.72 | 3.28 | 0.47 | 2.27 | 0.18 | 1.36 | **0.10** | **0.87** | 0.04 | 0.48 | 0.70 | 4.89 | **0.10** | **0.67** |
| PointButton2 | 0.17 | 2.17 | 0.53 | 6.04 | 0.50 | 2.19 | 0.11 | 2.07 | 0.07 | 1.15 | **0.13** | **0.81** | 0.63 | 5.58 | **0.14** | **0.87** |
| PointCircle1 | 0.86 | 6.19 | **0.45** | **0.00** | **0.51** | **0.12** | 0.38 | 1.10 | 0.52 | 5.89 | 0.46 | 0.85 | 0.64 | 3.11 | **0.46** | **0.93** |
| PointCircle2 | 0.86 | 9.80 | 0.69 | 9.40 | **0.50** | **0.48** | **0.53** | **0.85** | 0.81 | 9.42 | 0.48 | 0.55 | 0.50 | 1.55 | **0.51** | **0.94** |
| PointGoal1 | 0.55 | 1.91 | **0.40** | **0.92** | 0.46 | 0.57 | 0.38 | 0.48 | 0.68 | 1.55 | 0.33 | 0.68 | 0.78 | 1.65 | **0.51** | **0.71** |
| PointGoal2 | 0.40 | 1.81 | 0.58 | 2.75 | **0.57** | **0.93** | 0.47 | 1.09 | 0.34 | 0.73 | 0.26 | 0.64 | 0.77 | 6.22 | **0.53** | **0.74** |
| PointPush1 | 0.17 | 1.12 | **0.05** | **0.12** | 0.14 | 0.47 | 0.27 | 0.35 | **0.34** | **0.45** | 0.12 | 0.39 | 0.16 | 0.25 | **0.23** | **0.76** |
| PointPush2 | 0.04 | 1.58 | **0.06** | **0.23** | 0.16 | 1.23 | **0.27** | **0.46** | 0.22 | 1.59 | 0.08 | 0.48 | 0.16 | 1.74 | **0.05** | **0.33** |
| AntVel | 1.00 | 4.14 | **-1.01** | **0.00** | 0.94 | 2.27 | 0.29 | 0.05 | 0.90 | 0.00 | **0.95** | **0.72** | -1.01 | 0.00 | **0.91** | **0.00** |
| HalfCheetahVel | **0.69** | **0.00** | 0.56 | 0.80 | 0.57 | 1.93 | **0.92** | **0.97** | 0.90 | 0.00 | **0.92** | **0.77** | 0.83 | 1.00 | **0.83** | **0.00** |
| HopperVel | **0.07** | **0.91** | 0.19 | 2.76 | 0.23 | 0.42 | 0.08 | 2.10 | 0.18 | 0.87 | 0.25 | 0.26 | 0.08 | 0.83 | **0.65** | **0.15** |
| SwimmerVel | 0.67 | 11.58 | **0.09** | **0.77** | **0.68** | **0.87** | 0.51 | 1.05 | -0.04 | 0.51 | 0.56 | 3.35 | -0.08 | 0.01 | **0.49** | **0.34** |
| Walker2dVel | 0.33 | 3.80 | **0.07** | **0.00** | 0.50 | 1.54 | **0.09** | **0.18** | 0.66 | 0.37 | **0.81** | **0.68** | -0.01 | 0.00 | **0.10** | **0.34** |
| AntRun | **0.60** | **0.57** | 0.03 | 0.00 | **0.69** | **0.73** | 0.64 | 0.78 | 0.31 | 0.10 | **0.69** | **0.79** | 0.02 | 0.00 | **0.50** | **0.01** |
| BallRun | 0.61 | 2.05 | 0.30 | 0.74 | 0.28 | 0.35 | 0.66 | 1.41 | 0.22 | 0.03 | 0.22 | 1.38 | **0.33** | **0.93** | 0.32 | 0.85 |
| CarRun | 0.86 | 0.00 | 1.05 | 4.60 | 0.54 | 0.39 | 0.97 | 1.05 | 0.74 | 0.00 | 0.97 | 0.13 | 1.90 | 6.04 | **0.98** | **0.27** |
| DroneRun | 0.66 | 2.58 | 0.27 | 3.07 | **0.59** | **0.62** | 0.60 | 2.24 | 0.37 | 0.27 | 0.50 | 0.25 | 0.54 | 4.15 | 0.49 | 0.94 |
| AntCircle | 0.15 | 1.36 | **0.00** | **0.00** | 0.47 | 1.02 | 0.49 | 2.18 | 0.25 | 0.00 | **0.41** | **0.19** | 0.54 | 1.02 | 0.27 | 0.95 |
| BallCircle | 0.72 | 1.53 | **0.66** | **0.82** | 0.47 | 0.21 | **0.85** | **0.87** | 0.30 | 0.00 | 0.76 | 0.78 | 0.81 | 0.71 | 0.66 | 0.92 |
| CarCircle | 0.49 | 1.73 | **0.70** | **0.57** | 0.72 | 0.42 | 0.69 | 0.79 | 0.38 | 0.00 | 0.69 | 0.83 | **0.74** | **0.84** | 0.42 | 0.82 |
| DroneCircle | **0.23** | **0.26** | -0.23 | 0.33 | 0.53 | 0.45 | 0.63 | 1.56 | 0.48 | 0.00 | **0.55** | **0.83** | 0.25 | 0.94 | 0.24 | 0.89 |
| easysparse | 0.96 | 3.06 | **-0.06** | **0.05** | 0.05 | 0.03 | 0.06 | 0.35 | 0.31 | 0.05 | 0.00 | 0.16 | -0.05 | 0.05 | **0.68** | **0.12** |
| easymean | 0.67 | 2.27 | **-0.07** | **0.03** | 0.78 | 1.31 | -0.01 | 0.11 | 0.45 | 0.11 | 0.01 | 0.27 | -0.06 | 0.00 | **0.84** | **0.93** |
| easydense | 0.48 | 1.62 | **-0.06** | **0.03** | 0.50 | 1.35 | 0.00 | 0.10 | 0.55 | 0.60 | 0.09 | 0.17 | -0.06 | 0.03 | **0.81** | **0.67** |
| mediumsparse | 0.91 | 1.76 | **-0.09** | **0.03** | 0.96 | 1.56 | -0.08 | 0.03 | **0.44** | **0.00** | 0.27 | 0.03 | -0.08 | 0.05 | 0.30 | 0.10 |
| mediummean | 0.74 | 1.35 | **-0.08** | **0.03** | 0.72 | 1.19 | 0.19 | 0.42 | 0.51 | 0.07 | **0.84** | **0.97** | -0.06 | 0.00 | 0.26 | 0.19 |
| mediumdense | 0.90 | 1.57 | **-0.08** | **0.03** | 0.81 | 1.38 | 0.09 | 0.29 | 0.63 | 0.10 | **0.72** | **0.49** | -0.07 | 0.05 | 0.20 | 0.04 |
| hardsparse | 0.38 | 1.37 | **-0.05** | **0.03** | 0.29 | 0.88 | 0.11 | 0.35 | 0.33 | 0.14 | 0.52 | 0.54 | -0.05 | 0.03 | **0.53** | **0.34** |
| hardmean | 0.39 | 1.40 | **-0.05** | **0.05** | 0.11 | 0.28 | -0.04 | 0.03 | 0.25 | 0.01 | 0.24 | 0.10 | -0.05 | 0.03 | **0.45** | **0.02** |
| harddense | 0.31 | 1.06 | **-0.04** | **0.03** | 0.34 | 1.03 | -0.02 | 0.05 | 0.33 | 0.11 | 0.48 | 1.06 | -0.04 | 0.03 | **0.44** | **0.12** |
| **Safe/Optimal** | 8 | 0 | 23 | 0 | 19 | 6 | 24 | 6 | 31 | 4 | 33 | 8 | 20 | 2 | 38 | 15 |
| **Average** | 0.48 | 2.45 | 0.21 | 2.39 | 0.46 | 1.12 | **0.33** | **0.80** | 0.38 | 0.92 | 0.39 | 0.69 | 0.35 | 2.25 | **0.40** | **0.51** |

### C.3.2 SPARSE-STATE DENSE-ACTION TRAJECTORY

Our choice of the SS-DA representation is a deliberate design to balance maximizing rewards with ensuring safety.

- Sparse-State (SS): To capture long temporal abstractions, we build upon the hierarchical diffuser framework Li et al. (2023); Chen et al. (2024). Using sparse states (i.e., sampling states at intervals of $j$) allows the model to plan over a longer effective horizon. This encourages broader exploration and more strategic high-level decision-making compared to traditional dense-state representations, which can be myopic.

- Dense-Action (DA): While states are sparse, we maintain a dense action sequence. This ensures the continuity of the generated trajectory and allows the policy to specify fine-grained controls at every timestep. This dense control is vital for reacting to immediate state changes relevant to safety constraints, a capability that would be lost with sparse actions.

To demonstrate the benefits of our chosen representation ($j = 4$ by default) and its contribution to performance, we conducted an ablation study on the state sparsity interval $j$ in Table 8.

These results clearly show the performance trade-off and validate that our chosen $j = 4$ is the optimal setting for maximizing reward.

Table 5: Normalized DSRL Liu et al. (2023a) benchmark results. We compare our method (SDGD) with baseline methods BC and BCQ-Lag. ↑ means the higher the better. ↓ means the lower the better. Each value is averaged over 20 evaluation episodes and 3 seeds. Black: Unsafe agents. **Bold**: Safe agents whose normalized cost is smaller than 1. **Blue**: Safe agents with the highest reward.

| Task | BC | | BCQ-Lag | | SDGD (ours) | |
|---|---|---|---|---|---|---|
| | reward↑ | cost↓ | reward↑ | cost↓ | reward↑ | cost↓ |
| CarButton1 | -0.01 | 3.71 | 0.07 | 4.15 | **-0.03** | **0.63** |
| CarButton2 | -0.18 | 1.21 | 0.06 | 7.51 | **-0.05** | **0.52** |
| CarCircle1 | 0.74 | 17.70 | 0.71 | 17.67 | **0.14** | **0.76** |
| CarCircle2 | 0.81 | 28.15 | 0.73 | 22.23 | **0.34** | **0.15** |
| CarGoal1 | 0.31 | 1.62 | 0.29 | 3.84 | **0.31** | **0.63** |
| CarGoal2 | 0.24 | 3.25 | 0.29 | 6.55 | **0.10** | **0.58** |
| CarPush1 | 0.25 | 4.54 | 0.24 | 1.11 | **0.21** | **0.92** |
| CarPush2 | 0.17 | 4.05 | 0.14 | 2.76 | **0.04** | **0.70** |
| PointButton1 | 0.15 | 4.38 | 0.33 | 6.81 | **0.04** | **0.70** |
| PointButton2 | 0.20 | 4.45 | 0.34 | 8.18 | **0.05** | **0.92** |
| PointCircle1 | 0.76 | 13.25 | 0.34 | 3.63 | 0.29 | 1.49 |
| PointCircle2 | 0.65 | 14.15 | 0.62 | 8.53 | **0.49** | **0.56** |
| PointGoal1 | 0.58 | 2.64 | 0.63 | 3.54 | **0.43** | **0.86** |
| PointGoal2 | 0.62 | 10.32 | 0.63 | 10.47 | **0.21** | **0.89** |
| PointPush1 | 0.26 | 3.63 | 0.28 | 3.71 | **0.11** | **0.76** |
| PointPush2 | 0.09 | 5.15 | 0.06 | 2.29 | **0.05** | **0.81** |
| AntVel | 0.98 | 3.93 | 1.00 | 5.67 | **0.90** | **0.01** |
| HalfCheetahVel | 0.96 | 19.11 | 1.06 | 7.10 | **0.83** | **0.00** |
| HopperVel | 0.61 | 1.90 | 0.76 | 15.09 | **0.63** | **0.34** |
| SwimmerVel | 0.40 | 14.90 | 0.52 | 21.45 | **0.47** | **0.15** |
| Walker2dVel | 0.71 | 4.65 | **0.81** | **0.17** | **0.10** | **0.79** |
| AntRun | 0.72 | 5.27 | 0.68 | 6.06 | **0.50** | **0.04** |
| BallRun | 0.81 | 6.57 | 0.08 | 6.20 | **0.26** | **0.14** |
| CarRun | 0.97 | 0.36 | 0.98 | 2.64 | **0.97** | **0.03** |
| DroneRun | 0.67 | 4.45 | 0.73 | 11.45 | **0.35** | **0.58** |
| AntCircle | 0.63 | 11.32 | 0.52 | 4.98 | **0.20** | **0.98** |
| BallCircle | 0.68 | 6.45 | 0.69 | 3.69 | **0.49** | **0.81** |
| CarCircle | 0.62 | 5.73 | 0.57 | 2.98 | **0.40** | **0.90** |
| DroneCircle | 0.76 | 4.73 | 0.65 | 2.85 | 0.09 | 2.56 |
| easysparse | 0.32 | 4.73 | 0.99 | 8.83 | **0.62** | **0.02** |
| easymean | 0.22 | 2.68 | 1.00 | 5.45 | **0.60** | **0.39** |
| easydense | 0.20 | 1.70 | 0.19 | 1.24 | **0.71** | **0.82** |
| mediumsparse | 0.53 | 1.74 | **0.16** | **0.99** | **0.17** | **0.10** |
| mediummean | 0.66 | 2.94 | 0.40 | 1.81 | **0.19** | **0.11** |
| mediumdense | 0.65 | 3.79 | 0.37 | 2.50 | **0.20** | **0.11** |
| hardsparse | 0.28 | 1.98 | -0.02 | 0.22 | **0.44** | **0.32** |
| hardmean | 0.34 | 3.76 | 0.55 | 2.69 | **0.45** | **0.06** |
| harddense | 0.40 | 5.57 | 0.37 | 2.54 | **0.44** | **0.35** |

## C.4 FAILURE CASE ANALYSIS

This section provides a targeted analysis of one of the two failure cases, `DroneCircle` (Normalized Cost: 2.56), by performing a comparative analysis against a successful task from the same environment, `DroneRun` (Normalized Cost: 0.97). We find this failure is not due to a higher quantity of unsafe data, but rather to the qualitative structure of the dataset. The `DroneCircle` data is characterized by (1) highly fragmented, short-duration safe and unsafe segments, which (2) creates an ambiguous, non-separable safety boundary in the state space.

### C.4.1 TEMPORAL FRAGMENTATION OF SAFE/UNSAFE SEGMENTS

The data presented in Figure 9 reveals a critical difference in the temporal patterns of the two tasks. The `DroneCircle` dataset consists of trajectories that frequently "hop" in and out of the unsafe region.

- `DroneCircle` (FAIL): Trajectories exhibit very short bursts of unsafe behavior, with a mean violation length of only 12.1 steps. They also exit the safe state frequently, with the mean safe segment length being only 46.9 steps.

Table 6: Ablations on Two Kinds of Guidances Across Environments. Gray: Unsafe agents. **Bold**: Safe agents whose normalized cost is smaller than 1. Blue: Safe agents with the highest reward.

| Env | Task | w/o CFG reward↑ | w/o CFG cost↓ | w/o CG reward↑ | w/o CG cost↓ | Ours reward↑ | Ours cost↓ |
|---|---|---|---|---|---|---|---|
| SafetyGym | CarButton1 | -0.03 | 2.70 | **-0.06** | **0.60** | **-0.03** | **0.63** |
| | CarButton2 | 0.01 | 2.29 | **-0.05** | **0.52** | **-0.05** | **0.52** |
| | CarCircle1 | 0.58 | 18.86 | **-0.04** | **0.19** | **0.14** | **0.76** |
| | CarCircle2 | 0.69 | 19.67 | **0.24** | **0.28** | **0.34** | **0.15** |
| | CarGoal1 | 0.60 | 2.56 | **0.14** | **0.06** | **0.31** | **0.63** |
| | CarGoal2 | 0.52 | 8.86 | **-0.01** | **0.06** | **0.10** | **0.58** |
| | CarPush1 | 0.27 | 3.85 | **0.11** | **0.28** | **0.21** | **0.92** |
| | CarPush2 | 0.10 | 2.69 | **-0.01** | **0.23** | **0.04** | **0.58** |
| | PointButton1 | 0.34 | 8.62 | **-0.02** | **0.43** | **0.04** | **0.70** |
| | PointButton2 | 0.24 | 6.19 | **0.04** | **0.44** | **0.05** | **0.92** |
| | PointCircle1 | 0.43 | 26.21 | 0.29 | 1.49 | 0.29 | 1.49 |
| | PointCircle2 | 0.69 | 24.12 | **0.43** | **0.00** | **0.49** | **0.56** |
| | PointGoal1 | 0.58 | 2.62 | **0.26** | **0.63** | **0.43** | **0.86** |
| | PointGoal2 | 0.59 | 5.22 | **0.07** | **0.04** | **0.21** | **0.89** |
| | PointPush1 | 0.12 | 2.09 | **0.11** | **0.76** | **0.11** | **0.76** |
| | PointPush2 | 0.03 | 3.90 | **0.01** | **0.26** | **0.05** | **0.81** |
| | AntVel | 0.91 | 0.28 | **0.88** | **0.03** | **0.90** | **0.01** |
| | HalfCheetahVel | 0.33 | 0.00 | **0.42** | **0.00** | **0.83** | **0.00** |
| | HopperVel | 0.55 | 20.36 | **0.61** | **0.01** | **0.63** | **0.34** |
| | SwimmerVel | 0.46 | 0.80 | **0.23** | **0.15** | **0.47** | **0.15** |
| | Walker2dVel | 0.10 | 1.55 | **0.10** | **0.80** | **0.10** | **0.79** |
| | **Avg** | 0.41 | 8.17 | **0.20** | **0.51** | **0.28** | **0.65** |
| Bullet | AntRun | 0.35 | 0.56 | **0.24** | **0.08** | **0.50** | **0.04** |
| | BallRun | 0.42 | 5.45 | **0.13** | **0.00** | **0.26** | **0.14** |
| | CarRun | 0.98 | 1.88 | **0.76** | **0.00** | **0.97** | **0.03** |
| | DroneRun | 0.79 | 10.70 | **0.30** | **0.47** | **0.35** | **0.58** |
| | AntCircle | 0.54 | 14.52 | **0.12** | **0.67** | **0.20** | **0.98** |
| | BallCircle | 0.82 | 6.02 | **0.43** | **0.02** | **0.49** | **0.81** |
| | CarCircle | 0.81 | 7.96 | **0.42** | **0.86** | **0.40** | **0.90** |
| | DroneCircle | 0.52 | 10.39 | -0.09 | 2.56 | -0.09 | 2.56 |
| | **Avg** | 0.65 | 6.07 | **0.35** | **0.27** | **0.47** | **0.48** |
| MetaDrive | easysparse | 1.00 | 8.38 | **0.57** | **0.00** | **0.62** | **0.02** |
| | easymean | 1.00 | 7.46 | **0.56** | **0.00** | **0.60** | **0.39** |
| | easydense | 1.00 | 6.21 | **0.68** | **0.41** | **0.71** | **0.82** |
| | mediumsparse | 0.17 | 1.09 | **0.16** | **0.10** | **0.17** | **0.10** |
| | mediummean | 0.25 | 1.91 | **0.17** | **0.10** | **0.19** | **0.11** |
| | mediumdense | 0.18 | 1.05 | **0.18** | **0.10** | **0.20** | **0.11** |
| | hardsparse | 0.44 | 3.00 | **0.42** | **0.00** | **0.44** | **0.32** |
| | hardmean | 0.44 | 2.80 | **0.38** | **0.00** | **0.45** | **0.06** |
| | harddense | 0.44 | 3.05 | **0.35** | **0.04** | **0.44** | **0.35** |
| | **Avg** | 0.54 | 3.89 | **0.39** | **0.08** | **0.42** | **0.25** |

Table 7: Ablation: "CG-Cost / CFG-Reward" vs. SDGD (Ours). The variant consistently fails the safety constraint (Normalized Cost > 1.0).

| Baseline | CG-Cost / CFG-Reward | | SDGD (Ours) | |
|---|---|---|---|---|
| Task | Reward ↑ | Cost ↓ | Reward ↑ | Cost ↓ |
| CarButton1 | 0.06 | 2.01 | **-0.03** | **0.63** |
| CarButton2 | 0.02 | 1.11 | **-0.05** | **0.52** |
| HopperVel | 0.18 | 1.92 | **0.63** | **0.34** |
| CarRun | 2.18 | 3.84 | **0.97** | **0.03** |
| DroneRun | 0.77 | 9.38 | **0.35** | **0.58** |
| **Average** | 0.64 | 3.65 | **0.37** | **0.42** |

- `DroneRun` (SUCCESS): Trajectories show more "committed" behavior. Violation segments are 4.3 times longer (mean: 51.8 steps) , and safe segments are 2.4 times longer (mean: 112.6 steps).

---

**Algorithm 3** Sampling: "CG-Cost / CFG-Reward" Baseline

---

1: Given: Current state $s$, cost classifier $p_\phi$, CFG strength $w$, CG strength $s$.
2: $\tau_T \sim \mathcal{N}(\mathbf{0}, \mathbf{I}); z_0 = $ cost limit
3: **for** $t = 0, 1, ..., T$ **do**
4:   **for** $i = N, ..., 1$ **do**
5:     $\nabla_{\tau_i} C(\tau_i) \leftarrow p_\phi(C|\tau_i)$                            ▷ Get cost gradient
6:     $\hat{\epsilon}_i = (1 + w)\epsilon_\theta(\tau_i, R) - w\epsilon_\theta(\tau_i)$              ▷ CFG for Reward
7:     $\mu_\theta(\tau_{i-1}) \leftarrow \text{Denoise}(\tau_i, \hat{\epsilon}_i)$         ▷ Get base mean from reward guidance
8:     **if** $C(\tau_i) > z_t$ **then**               ▷ **Apply CG-Cost if predicted unsafe**
9:       $\tau_{i-1} \sim \mathcal{N}(\mu_\theta(\tau_{i-1}) - s\nabla_{\tau_i} C(\tau_i), \Sigma_\theta(\tau_{i-1}))$
10:     **else**                    ▷ Otherwise, only use reward-guided mean
11:       $\tau_{i-1} \sim \mathcal{N}(\mu_\theta(\tau_{i-1}), \Sigma_\theta(\tau_{i-1}))$
12:     **end if**
13:   **end for**
14:   Execute first action of trajectory and get cost $c_t$, $z_{t+1} = (z_t - c_t)/\gamma$
15: **end for**

---

Table 8: Ablations on sparse-state and dense-action in Bullet-Safety-Gym. Blue-shaded cells indicate optimal values per metric category. Gray: Unsafe agents.

| Task | j=1 reward↑ | cost↓ | j=2 reward↑ | cost↓ | j=4 reward↑ | cost↓ | j=8 reward↑ | cost↓ | j=16 reward↑ | cost↓ |
|------|---|---|---|---|---|---|---|---|---|---|
| AntRun | 0.36 | 0.00 | 0.42 | 0.04 | 0.50 | 0.04 | 0.31 | 0.05 | 0.21 | 0.03 |
| BallRun | 0.24 | 0.00 | 0.26 | 0.00 | 0.26 | 0.14 | 0.28 | 0.65 | 0.25 | 0.44 |
| CarRun | 0.91 | 0.02 | 0.96 | 0.16 | 0.97 | 0.03 | 0.98 | 0.34 | 0.91 | 0.26 |
| AntCircle | 0.13 | 0.79 | 0.24 | 0.94 | 0.20 | 0.98 | 0.15 | 0.85 | 0.03 | 0.96 |
| BallCircle | 0.31 | 0.18 | 0.39 | 0.29 | 0.49 | 0.81 | 0.45 | 0.22 | 0.58 | 0.41 |
| CarCircle | 0.32 | 0.88 | 0.47 | 0.90 | 0.40 | 0.90 | 0.54 | 0.93 | 0.54 | 1.37 |
| **Avg** | 0.38 | 0.31 | 0.46 | 0.39 | 0.47 | 0.48 | 0.44 | 0.51 | 0.42 | 0.58 |

This temporal fragmentation in `DroneCircle` makes it extremely difficult for the model to learn a coherent, long-horizon safe policy, as the dataset lacks sufficient examples of sustained safe trajectories.

### C.4.2 SPATIALLY AMBIGUOUS SAFETY BOUNDARY

This temporal fragmentation directly results in the spatial ambiguity observed in the PCA projection (Figure 10).

- As seen in Figure 10 (Left), the frequent temporal hopping in `DroneCircle` causes the safe (blue) and unsafe (red) states to be highly intermingled. This creates an ambiguous,

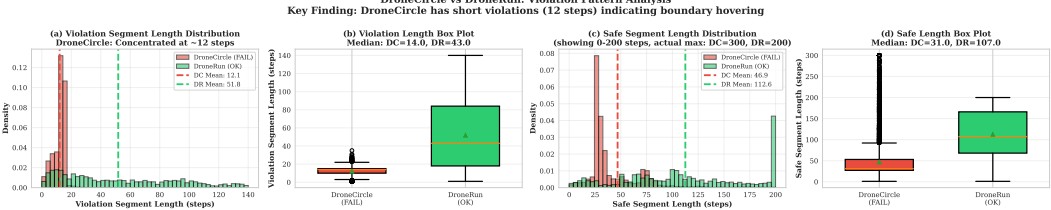

Figure 9: Analysis of temporal segment lengths for `DroneCircle` (FAIL) vs. `DroneRun` (OK). (a, b) `DroneCircle` violations are brief (mean 12.1 steps), while `DroneRun` violations are sustained (mean 51.8). (c, d) Similarly, `DroneCircle` safe segments are fragmented (mean 46.9 steps), unlike the long safe segments in `DroneRun` (mean 112.6). This temporal fragmentation leads to the ambiguous spatial boundary seen in Figure 10.

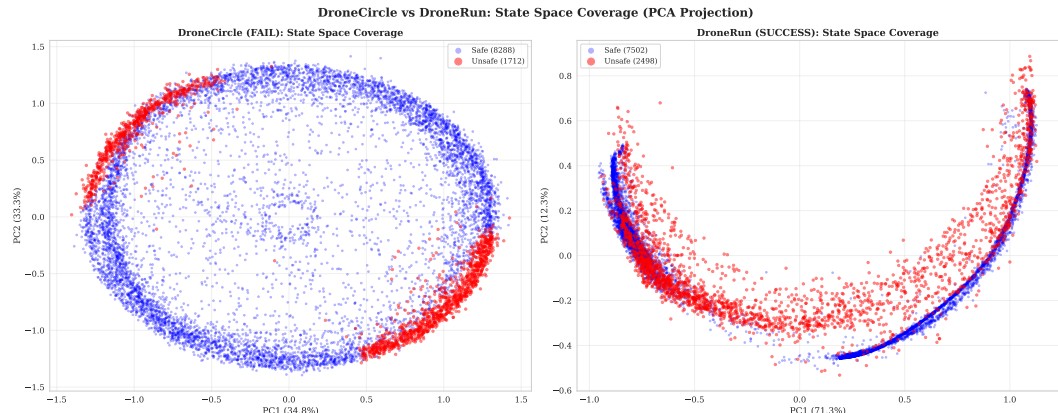

Figure 10: PCA comparison of state space distributions. (Left) `DroneCircle` (FAIL) shows highly intermingled safe (blue) and unsafe (red) states, creating an ambiguous boundary. (Right) `DroneRun` (SUCCESS) exhibits clear spatial separation, forming a distinct, learnable boundary.

non-separable safety boundary that is inherently difficult for our guidance mechanisms (both CFG and CG) to learn.

- Conversely, the long, distinct temporal segments in `DroneRun` (Figure 10 Right) create clear spatial separation. The unsafe states are clustered, forming a clean, learnable boundary that the model can easily identify and avoid.

In conclusion, the `DroneCircle` failure is a data-driven issue. The dataset's structure, defined by sporadic, short-burst violations, creates a spatially "polluted" and non-separable state space. This makes it challenging for the model to learn a complete and sustained safe trajectory, confirming our hypothesis that dataset quality and structure are fundamental challenges for offline safe RL.

### C.5 TASK DESCRIPTION

DSRL Liu et al. (2023a) collects 3 benchmarks (Safety-Gymnasium Ji et al. (2023), Bullet-Safety-Gym Gronauer (2022) and MetaDrive Li et al. (2022)) over 75000 diverse trajectories from 38 tasks.

**Safety-Gymnasium.** Its backend is Mujoco, and it has two types of environments (Velocity and others). Velocity includes 5 tasks, Ant, HalfCheetah, Hopper, Swimmer, Walker2d, and a total of 11,399 trajectories were collected. The other type of environment includes five types of environments (Goal, Button, Push, Circle) and two types of agents (point and car), a total of 16 tasks, and 40,310 trajectories were collected.

**Bullet-Safety-Gym.** Its backend is PyBullet, and it has two types of environments (Run and Circle) with four types of agents (Ball, Car, Drone and Ant), a total of 8 tasks, and 14498 trajectories.

**MetaDrive.** Its backend is Panda3D, and it has a driving environment with three types of roads (easy, medium and hard) with three types of traffics (sparse, mean, dense), a total of 9 tasks, and 9000 trajectories.

The details of the state space and action space of each environment are shown in Table 9, and its visualization is shown in Figure 11.

**Evaluation Metrics.** We adopt the normalized reward $R_{\text{normalized}} := \frac{R_\pi - r_{\min}}{r_{\max} - r_{\min}}$ and the normalized cost $C_{\text{normalized}} = \frac{C_\pi + \epsilon}{l + \epsilon}$, as the comparison metrics Fu et al. (2020); Liu et al. (2023b), where $R_\pi = \sum_t r_t$ is the policy's reward return and $C_\pi = \sum_t c_t$ is its cost return, $r_{\min}$ and $r_{\max}$ are the minimum and maximum rewards in the task, and $l$ is the cost threshold which is set by users.

Table 9: Dataset details.

| Benchmarks | Task | Max Timestep | Action Space | State Space | Max Cost |
|---|---|---|---|---|---|
| SafetyGymnasium | CarButton1 | 1000 | 2 | 88 | 250 |
| | CarButton2 | 1000 | 2 | 88 | 300 |
| | CarCircle1 | 500 | 2 | 40 | 250 |
| | CarCircle2 | 500 | 2 | 40 | 400 |
| | CarGoal1 | 1000 | 2 | 72 | 120 |
| | CarGoal2 | 1000 | 2 | 72 | 200 |
| | CarPush1 | 1000 | 2 | 88 | 200 |
| | CarPush2 | 1000 | 2 | 88 | 250 |
| | PointButton1 | 1000 | 2 | 76 | 200 |
| | PointButton2 | 1000 | 2 | 76 | 250 |
| | PointCircle1 | 500 | 2 | 28 | 200 |
| | PointCircle2 | 500 | 2 | 28 | 300 |
| | PointGoal1 | 1000 | 2 | 60 | 100 |
| | PointGoal2 | 1000 | 2 | 60 | 200 |
| | PointPush1 | 1000 | 2 | 76 | 150 |
| | PointPush2 | 1000 | 2 | 76 | 200 |
| | Swimmer | 1000 | 2 | 8 | 200 |
| | Hopper | 1000 | 3 | 11 | 250 |
| | HalfCheetah | 1000 | 6 | 17 | 250 |
| | Walker2d | 1000 | 6 | 17 | 300 |
| | Ant | 1000 | 8 | 27 | 250 |
| BulletSafetyGym | AntRun | 200 | 8 | 33 | 150 |
| | BallRun | 100 | 2 | 7 | 80 |
| | CarRun | 200 | 2 | 7 | 40 |
| | DroneRun | 200 | 4 | 17 | 140 |
| | AntCircle | 500 | 8 | 34 | 200 |
| | BallCircle | 200 | 2 | 8 | 80 |
| | CarCircle | 300 | 2 | 8 | 100 |
| | DroneCircle | 300 | 4 | 18 | 100 |
| MetaDrive | easysparse | 1000 | 2 | 261 | 85 |
| | easymean | 1000 | 2 | 261 | 85 |
| | easydense | 1000 | 2 | 261 | 85 |
| | mediummean | 1000 | 2 | 261 | 50 |
| | mediumsparse | 1000 | 2 | 261 | 50 |
| | mediumdense | 1000 | 2 | 261 | 50 |
| | hardsparse | 1000 | 2 | 261 | 85 |
| | hardmean | 1000 | 2 | 261 | 85 |
| | harddense | 1000 | 2 | 261 | 85 |

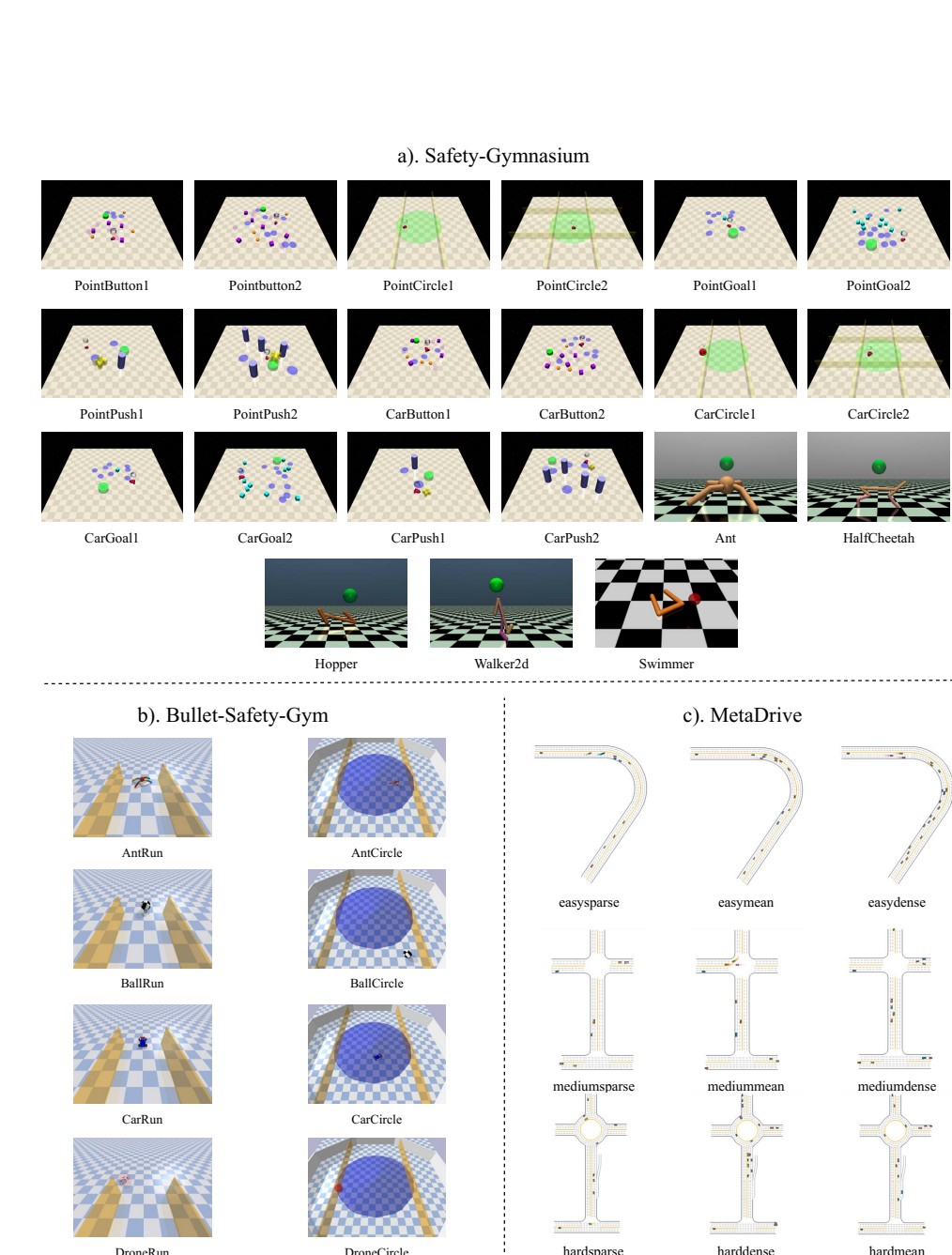

Figure 11: **Visualization of the three environments.** All evaluation tasks are shown (21 tasks are from Safety-gymnasium, 8 tasks are from Bullet-Safety-Gym, and 9 tasks are from MetaDrive.)

## D  A DETAILED DISCUSSION ON THE LIMITATIONS

Despite the promising results, the current work possesses certain limitations that warrant further investigation.

**Hyperparameter Sensitivity and Selection.** The efficacy of the diffusion model for trajectory generation is notably dependent on a considerable number of hyperparameters. These include, but are not limited to, the number of denoising steps, the predetermined length of the generated trajectory, the guidance scale $s$ for classifier guidance, and the conditional weight $w$ for classifier-free guidance. The specific choices for these parameters can significantly influence the qualitative and quantitative aspects of the final generated trajectories. A key limitation of the current study is the absence of a methodology for automated, context-aware hyperparameter selection. Developing mechanisms that enable the model to dynamically adapt or select optimal hyperparameters based on the specific task or environmental context remains an open research question.

**Real-World Deployment Constraints.** A primary objective for practical application involves deploying the proposed algorithms on resource-constrained edge devices for real-time trajectory generation. This ambition introduces substantial challenges. The computational demands inherent in diffusion models, particularly for iterative denoising processes, can be prohibitive for typical edge computing hardware, which often operates under strict limitations regarding processing power, memory, and energy consumption. Consequently, these hardware requirements currently restrict the range of feasible application scenarios and hinder widespread adoption in real-world systems where immediate, on-device trajectory synthesis is critical. Overcoming these deployment hurdles is essential for translating the theoretical benefits of the model into tangible, real-time solutions.

## E  USE OF LARGE LANGUAGE MODELS (LLMs)

A large language model (LLM) was used as a general-purpose writing assistant for tasks such as improving grammar, clarity, and style throughout the manuscript. In all instances, the authors reviewed, revised, and edited the generated text to ensure it accurately reflects the research. The authors take full responsibility for all content, including its scientific accuracy and integrity, presented in this submission.

