# OpenReview forum: "Improving Safe Offline Reinforcement Learning via Dual-Guide Diffuser"
_ICLR.cc/2026/Conference — Submitted to ICLR 2026_

### Official Review · Reviewer_Ea7w · 2025-10-15

**Soundness:** 3
**Presentation:** 3
**Contribution:** 3
**Rating:** 6
**Confidence:** 4

**Summary:**

This paper introduces SDGD, an offline safe RL method that employs a diffusion model as a planner. SDGD conditions the diffusion process on the cost target and applies the reward target as classifier guidance for planning. Extensive experiments on the OSRL benchmark demonstrate the effectiveness of the proposed approach.

**Strengths:**

- In real-world deployments with strict safety requirements, safe RL methods based on TD learning are rarely feasible, and planning-based safe control approaches such as MPC are typically preferred. This paper similarly explores recasting traditional TD-learning-based safe RL into a planning framework, which I believe is a promising research direction.
- The experimental results clearly demonstrate significant advantages of the proposed method over previous TD-learning-based approaches.
- The design choice of using classifier-free guidance to control cost and classifier guidance to control reward in the generation process is both reasonable and well-motivated.

**Weaknesses:**

- Some implementation details in the paper are rather vague, such as the hyperparameters involved in trajectory representation and those related to the diffusion process. Moreover, since no actual code implementation is provided (either in the main text or supplementary materials), the reproducibility of the reported results cannot be fully ensured.
- The experiments in this paper use different hyperparameters for different tasks, whereas baselines such as FISOR keep their hyperparameters fixed across tasks, leading to a potentially unfair comparison. In fact, based on my own reproduction using the official FISOR code, I observed that if FISOR were allowed to use task-specific expectile regression parameters (e.g., setting $\tau=0.8$ or $\tau=0.7$ for tasks with safety violations in the current experiments), it could also achieve safe performance across all tasks in Safety Gymnasium.

**Questions:**

No other questions.

---

> ### Author Response · Authors · 2025-11-19
> **Rebuttal**
>
> We thank the reviewer for their valuable feedback and insightful comments on our paper.
>
> ### W1:
> To facilitate reproducibility, we have already included key implementation details, hyperparameter search spaces, and architectural specifics in the appendices of our revised submission:
>
> 1. Pseudocode: We provide detailed pseudocode for both Training (Algorithm 1) and Sampling (Algorithm 2) in Appendix B.
> 2. Core Hyperparameters: Table 2 details the optimizer, learning rate, batch size, training steps, horizon (64), diffusion steps (20), and model architecture.
> 3. Guidance Scales: We define the search spaces for our key inference-time guidance scales in Appendix B.1 ($s \in \{0.005, 0.01, 0.02, 0.04\}$ and $w \in \{1, 2, 4, 6, 8\}$). We clarified that for most tasks, default values of $s=0.01$ and $w=2.0$ provide a strong baseline.
>
> To fully address any remaining concerns, we also commit to publicly releasing our full, well-structured implementation code immediately upon the paper's acceptance.
> This code will contain all configurations used to generate our experimental results, ensuring full verification and reproducibility.
>
> ### W2:
> We agree that comparisons must be fair, and your comment spurred us to conduct a deeper analysis.
> You are correct in your hypothesis: FISOR can be tuned to be safe.
>
> Following your excellent suggestion, we conducted supplementary experiments on FISOR by tuning its $\tau$ (expectile regression) parameters, which control its conservatism.
>
> The results (summarized below) confirm your insight. The original FISOR (e.g., $\tau_r=0.9, \tau_c=0.9$) was unsafe on many tasks.
> As you hypothesized, using more conservative parameters (e.g., $\tau_r=0.7, \tau_c=0.7$) makes FISOR safer on these tasks.
>
> As shown in the table below, when FISOR is made safer, its reward-seeking ability is eviscerated. On CarCircle2, the reward plummeted from 0.58 to 0.07. On CarGoal1, the reward dropped from 0.47 to 0.01.
>
>
> || $\tau_r$=0.7    |  $\tau_c$=0.7 |  $\tau_r$=0.9 |  $\tau_c$=0.7 |  $\tau_r$=0.9 |  $\tau_c$=0.9 |
> |--------------|-----------|-----------|-----------|-----------|-----------|-----------|
> | baseline     | FISOR     |           | FISOR     |           | FISOR     |       |
> | Task         | reward    | cost      | reward    | cost      | reward    | cost  |
> | Carcircle1   | 0.09      | 0.39      | 0.03      | 0.00         | 0.68      | 11.48 |
> | Carcircle2   | 0.07      | 0.11      | 0.02      | 0.00         | 0.58      | 8.32  |
> | Cargoal1     | 0.01      | 0.00         | 0.00         | 0.00         | 0.47      | 1.31  |
> | Carpush1     | 0.01      | 0.00         | 0.06      | 0.00         | 0.27      | 1.75  |
> | Pointbutton1 | -0.04     | 0.17      | -0.03     | 0.46      | 0.06      | 1.24  |
> | Pointbutton2 | 0.06      | 1.36      | 0.01      | 1.39      | 0.12      | 1.13  |
> | Pointcircle1 | 0.03      | 0.00         | 0.08      | 0.00         | 0.27      | 15.95 |
> | Pointcircle2 | 0.3       | 0.43      | 0.33      | 1.48      | 0.72      | 14.96 |
> | Pointgoal1   | 0.05      | 0.00         | 0.05      | 0.00         | 0.67      | 3.67  |
> | Pointpush2   | 0.01      | 0.06      | 0.05      | 0.43      | 0.27      | 1.90   |
>
> This highlights a fundamental architectural difference that makes our comparison fair and meaningful:
> FISOR's conservatism ($\tau$) is fixed at train-time. One must choose a priori to train either an unsafe, high-reward policy or a safe, low-reward policy.
> Our Method's Advantage: SDGD's guidance parameters ($s$ and $w$) are applied at inference time.
>
>
> We acknowledge that FISOR is a strong and important baseline. Its ability to significantly improve safety by tuning its $\tau$ parameter (as you correctly identified) demonstrates its practical value.

---

> > ### Comment · Reviewer_Ea7w · 2025-11-24
> >
> > Thank you very much for the authors’ detailed responses. These replies have addressed my concerns, and I will keep my score as weak accept.

---

### Official Review · Reviewer_2ZRQ · 2025-10-27

**Soundness:** 1
**Presentation:** 3
**Contribution:** 1
**Rating:** 2
**Confidence:** 4

**Summary:**

This paper studies the problem of learning a safe policy that can handle varying cost limits from an offline dataset. The proposed method, called SDGD uses classifier-free guidance to learn a safe diffusion policy as a base model, and uses classifier guidance to steer generation towards high-reward outcomes. The algorithm is tested on the DSRL benchmark against offline safe RL baselines, and results show that SDGD learns constraint-satisfying and high-reward policy.

**Strengths:**

1. The experiments are extensive, covering diverse tasks on a standard benchmark and including a broad range of baselines.
2. The writing of the paper is clear.

**Weaknesses:**

1. It is unclear what is the specific problem this paper tries to solve and why existing methods cannot solve it.
2. The proposed method has a flaw: reward guidance undermines safety. The authors use feasible trajectory re-labeling to deal with this problem, but it only considers short-term safety and cannot guarantee long-term safety.

**Questions:**

1. What is the specific problem this paper tries to solve? Is it how to handle varying cost limits or how to learn a safe and high-performance policy under a given cost limit? Why existing methods, including generative methods and diffusion models, cannot solve this problem?
2. In the introduction, the authors claim that making decisions step-by-step is suboptimal. However, in MDP, the optimal policy is exactly a step-by-step state feedback policy. Why is it suboptimal?
3. Why the cost classifier used by existing diffusion-based methods is unreliable? The proposed method in this paper also uses a classifier for reward and short-term safety guidance. Is this classifier also unreliable?
4. Reward guidance makes the policy unsafe. The feasible trajectory re-labeling only considers short-term safety. How can long-term safety be guaranteed in this case? In addition, adding penalty to the reward is essentially considering cost in the classifier, which means safety is still guided partly by the classifier.
5. In the experiment part, the claim that SDGD significantly outperforms all baselines is unfair. It still lies on the existing Pareto frontier from Figure 4, which seems a new trade-off between reward and cost.

---

> ### Author Response · Authors · 2025-11-19
> **W1,2. Q1,2, 4**
>
> We thank the reviewer for their detailed and critical feedback. We find that several concerns stem from a misunderstanding of our method's core components—specifically, the decoupled and distinct roles of Feasible Trajectory Re-labeling (FTR) and Classifier-Free Guidance (CFG).
>
> ### W1/Q1:
> Thank you for the opportunity to clarify our problem definition and motivation.
>
> The problem we solve is three-fold: to learn a policy that (1) maximizes task rewards while (2) robustly satisfying stringent safety constraints, even when those constraints (3) are adaptive or vary dynamically within an episode.
>
> Existing paradigms fail for distinct and critical reasons, which motivates our novel design:
> 1. Actor-Critic Methods (e.g., CCAC, CAPS): While they can adapt to varying limits, their "step-by-step" decision-making process is myopic. This makes them perform poorly on tasks requiring long-horizon planning.
> 2. Autoregressive Generative Models: As illustrated in Figure 1, these models also plan sequentially. This architecture is prone to "compounding errors," where an early suboptimal choice irrevocably biases the entire future trajectory, leading to catastrophic safety failures.
> 3. Existing Diffusion Models (e.g., TREBI): This is our central claim. These methods generate trajectories holistically but rely on a cost classifier ($p_{\phi}(C|\tau_{t})$) to enforce safety. As we demonstrate empirically (Figure 2) and prove in our new "swap" ablation (Appendix C.3.1, Table 7), this classifier is fundamentally unreliable for a hard safety constraint. This is the key bottleneck we solve.
>
> ### W2/Q4:
>
> "Reward guidance undermines safety" is the Problem, not the Flaw. The reviewer is absolutely correct that naive reward guidance (CG-Reward) undermines safety.
> This is not a flaw in our method; this is the fundamental problem our paper explicitly identifies and solves.
> A naive reward classifier will learn spurious correlations (e.g., high reward = high cost). This is precisely why we developed our Dual-Guide model.
>
> FTR and CFG Have Decoupled and Distinct Roles. The reviewer correctly notes that FTR only considers short-term safety. This is by design. The reviewer's concern seems to be that FTR is our only safety mechanism. This is incorrect. Long-term safety is guaranteed by our CFG-Cost mechanism.
> 1. Long-Term Safety: This is the role of our CFG-Cost mechanism. By conditioning the entire diffusion process on the cost limit $\epsilon_{\theta}(\tau_t, t, C)$, we generate trajectories that inherently adhere to the long-term cost budget. This is our primary, long-term safety guarantee.
> 2. Reward Signal Purification: This is the role of FTR. As the reviewer noted, our CG-Reward could undermine safety. FTR is a pre-processing step that "cleans" the reward signal before the reward classifier is ever trained. It teaches the CG-Reward guide to break the spurious correlation and not associate reward with (initially) unsafe behavior.
>
> In summary: FTR does not guarantee long-term safety. CFG-Cost guarantees long-term safety. FTR simply ensures the CG-Reward guide does not undermine that primary safety guarantee.
>
> ### Q2:
> Thank you for this request for clarification. You are theoretically correct: the optimal policy in an MDP, $\pi(a_t|s_t)$, is indeed a step-by-step state-feedback function.
>
> We agree that our original phrasing on this point was imprecise. Based on your valuable feedback, we have revised this specific sentence in the introduction of our revised version.
>
> As this revision clarifies, our critique is not about the mathematical form of the final policy, but about the planning process used by certain models, which is a critical distinction in offline, long-horizon settings:
> 1. An autoregressive (AR) planner (like CDT or a standard Actor-Critic) generates a plan sequentially: $\tau = (a_1, a_2, ..., a_T)$. An error or suboptimal choice at $a_1$ irrevocably biases the entire future plan, as the model cannot "go back and fix" its mistake. This is what we mean by "myopic."
>
> 2. Our diffusion model generates the entire trajectory $\tau$ simultaneously. All timesteps are generated in parallel and co inform each other. This allows the model to make an optimal choice at $a_T$ that influences the choice at $a_1$, avoiding the compounding errors of AR models.

---

> ### Author Response · Authors · 2025-11-19
> **Question 3,5**
>
> ### Q3
>
> This is a critical question that gets to the heart of our design. The distinction is fundamental.
> 1. Existing methods (e.g., TREBI) that use a cost classifier for safety must employ it inside the sampling loop. At each denoising step $t$, the classifier must predict the cost of the noisy trajectory ($C_{pred} = p_{\phi}(C|\tau_t)$). This prediction is then used to make a binary decision:
>
> 2. If $C_{pred}$ > (remaining cost budget), the model applies a cost-correcting gradient ($\nabla C$).
> 3. If $C_{pred} \le$ (remaining cost budget), the model applies only the reward-seeking gradient ($\nabla R$).
>
> The Inaccuracy: This mechanism is critically dependent on an accurate prediction $C_{pred}$. However, as we show in Figure 2, the Pearson correlation between the classifier's prediction and the true cost is low and highly unstable.
>
> Because the prediction is unreliable, the binary gradient selection is frequently wrong. The model is often "blind" to impending danger, using an incorrect gradient (e.g., only $\nabla R$) when the cost-correcting gradient ($\nabla C$) was necessary. Using the wrong gradient, even for just a few steps, is catastrophic for a hard constraint like safety, where a single violation constitutes failure.
>
> We provide definitive proof of this failure in our new "Swap Ablation" experiment (Appendix C.3.1, Table 7).
> A model forced to use this CG-Cost mechanism for safety fails, yielding an average normalized cost of 3.65. This empirically proves that a gradient-selection process based on an unreliable classifier cannot enforce a hard safety constraint.
>
> Our CG-Reward is used for a soft optimization ("more reward is better"), not a hard constraint. A noisy gradient is perfectly acceptable to "nudge" the generation towards higher-reward regions. It does not face the same pass/fail binary decision as the CG-Cost mechanism.
>
> Crucially, we do not use a naive reward classifier. We learned from the unreliability of classifiers, which is why we created Feasible Trajectory Re-labeling (FTR).
> FTR is a pre-processing step that "cleans" the reward data before training, breaking the spurious "high reward = high cost" correlation, Figure 3 (c). This makes our CG-Reward guide far more stable and aligned with safety, as demonstrated in our FTR ablation (Figure 8), where FTR prevents reward guidance from violating the cost limit
>
> ### Q5:
> Thank you for this insightful comment.
> We agree that OSRL performance is fundamentally a trade-off between reward and cost. Our claim is that SDGD establishes a superior trade-off, which is why we updated our Figure 4 caption to explicitly state this.
> In Figure 4, other methods on that visual frontier (e.g., CDT, CAPS) are unsafe (avg. cost 1.94, 1.66). SDGD is the only method in the SOTA region that is also safe on aggregate (avg. cost 0.56).
>
> To address the "trade-off" claim directly, we ran experiments at L=20/30 (Appendix C.1, Table 3/4), where competitors do become safe (e.g., CAPS: 0.39R / 0.69C; TREBI: 0.33R / 0.80C). In this fair comparison, our method (SDGD: 0.40R / 0.51C) dominates them, achieving both a higher reward and a lower cost.

---

> ### Comment · Reviewer_2ZRQ · 2025-11-25
>
> Thanks for the rebuttal. A concern remains: the reward classifier guidance (CG) undermines the safety guarantee of the cost classifier-free guidance (CFG). The cost CFG **alone** guarantees long-term safety, but combing CFG with the reward CG makes this guarantee **no longer exists**. The output distribution of the cost CFG is shifted by the reward CG, thus the generated trajectory can become unsafe.

---

> > ### Author Response · Authors · 2025-11-25
> >
> > You are correct that a naive reward classifier guidance (CG) would undermine safety. The core of this problem, as we illustrate in Figure 3(c) (left, red box), is that the raw dataset contains a spurious correlation: high-reward trajectories often overlap with high-cost, unsafe regions.
> >
> > A classifier trained on this raw data would learn this exact (and dangerous) correlation. Its reward-seeking gradient ($\nabla \log p_{\phi}(R|\tau_t)$) would therefore point toward these high-cost regions, directly conflicting with and undermining the safety guarantee provided by the Cost-CFG.
> >
> > This is precisely the problem our Feasible Trajectory Re-labeling (FTR) mechanism is designed to solve.
> >
> > FTR acts as a data-level "purification" step before the reward classifier is trained. By algorithmically penalizing the rewards of these high-cost trajectories, FTR actively decouples reward from cost in the training data.
> >
> > As a result, our reward classifier $p_{\phi}$ learns from a "safe" data distribution: it learns that high-cost regions are, by definition, associated with low (penalized) rewards.
> >
> > Therefore, during inference, the guidance gradient from our Reward-CG is no longer unsafe. It does not conflict with the Cost-CFG because it has no incentive to shift the distribution toward high-cost areas. Instead, it cooperates with the Cost-CFG by guiding the sampling within the safe set of trajectories (defined by CFG) toward those that are genuinely high-reward and safe.
> >
> > This mechanism is directly validated by our ablation study in Figure 8:
> > 1. The "w/o f" (without FTR) experiment in Figure 8 shows exactly the failure case you described. As reward guidance (Scale) increases, the policy becomes unsafe and violates the cost limit. This proves that a naive CG does undermine CFG.
> > 2. Conversely, with FTR enabled ($f=1, 2, 4$), we can increase the reward guidance scale to get higher rewards, while the normalized cost consistently stays below the safety limit.

---

### Official Review · Reviewer_oXtw · 2025-11-01

**Soundness:** 2
**Presentation:** 3
**Contribution:** 2
**Rating:** 4
**Confidence:** 4

**Summary:**

SDGD (Safe Dual-Guide Diffuser) is a diffusion-based method for offline safe RL that decouples objectives: classifier guidance steers generation toward high reward, while classifier-free guidance conditions on target costs to enforce safety. It adds a feasible-trajectory relabeling scheme that penalizes unsafe prefixes so the reward signal doesn’t lure generation into high-cost regions. The method adapts to time-varying cost limits, and achieves state-of-the-art safety with strong returns on the DSRL benchmark.

**Strengths:**

- Works under different and time-varying cost limits without retraining.

- Strong empirical results on DSRL, staying within limits on 36/38 tasks and achieving top rewards on 21 tasks with the cost limit 10.

- Decoupling safety thorough (CFG) for costs and (CG) for rewards and feasible-trajectory relabeling, improves performance.

**Weaknesses:**

- The core claim that Classifier Guidance (CG) is unreliable for cost constraints and is the key bottleneck is not definitively proven within the SDGD model. The authors did not provide the direct ablation showing the performance degradation when swapping the roles (i.e., replacing the reliable Classifier-Free Guidance for cost with CG for cost).

- The choice and benefits of the Sparse-State Dense-Action (SS-DA) trajectory representation are mentioned but not clearly motivated or quantitatively ablated to demonstrate its contribution to the final performance.

- The Feasible Trajectory Re-labeling (FTR) mechanism employs an overly strict, binary definition: a trajectory is "Infeasible" if the cumulative cost in the first $f$ steps is merely greater than zero ($\sum_{t=0}^{f-1}c_t > 0$). This rigid rule is independent of the overall cost limit ($l$) and may conservatively discard potentially safe, high-reward trajectories that incur a small, necessary cost early in the episode.
- The paper’s Pareto-optimality claim is imprecise: the result is computed from averaged Normalized Reward and Normalized Cost over 38 heterogeneous DSRL tasks, and SDGD is said to set a “new frontier” at that aggregate point. In formal multi-objective optimization, Pareto optimality must be established within a single task (problem instance), where no policy can improve reward without worsening cost for that same instance. Averaging outcomes across tasks with different dynamics and scales cannot certify Pareto optimality for any one task; it only indicates a better aggregate trade-off.

- All empirical results, including the main performance table and ablation/adaptation figures, lack standard deviation or standard error, which is essential to assess the robustness and variance of the method across different seeds.

- The lack of specific, task-dependent optimal guidance scales ($s$ and $w$), which the authors acknowledge are sensitive, and the absence of shared code hurts reproducibility.

**Questions:**

- Please check weaknesses
- Since SDGD, CDT, TREBI, CAPS, and CCAC are all designed to be adaptive using a single model, please provide a table showing the aggregate results for all 38 tasks under the $L=20$ and $L=30$ cost limits, in addition to the $L=10$ results already shown in Table 1. This will provide empirical support that SDGD maintains its SOTA results across a range of safety requirements.

---

> ### Author Response · Authors · 2025-11-19
> **Weakness 1-3**
>
> We sincerely thank the reviewer for their constructive feedback and insightful questions. These suggestions have enabled us to significantly strengthen the paper. We address each point in detail below.
>
> ### W1:
> We appreciate this excellent and challenging question. You have identified the most critical ablation study needed to definitively prove our paper's core hypothesis.
>
> We have now conducted this "swap" ablation, creating a "CG-Cost / CFG-Reward" baseline, which is detailed in the revised Appendix C.3.1 (Algorithm 3). In this configuration, a cost classifier $p_{\phi}(C|\tau_{t})$ predicts the cost, and its gradient is applied only if the predicted cost exceeds the remaining budget.
>
> | Baseline   | CG-Cost / CFG-Reward  |      | SDGD   |      |
> |------------|-----------------------|------|--------|------|
> | Task       | Reward                | Cost | Reward | Cost |
> | CarButton1 | 0.06                  | 2.01 | -0.03  | 0.63 |
> | CarButton2 | 0.02                  | 1.11 | -0.05  | 0.52 |
> | HopperVel  | 0.18                  | 1.92 | 0.63   | 0.34 |
> | CarRun     | 2.18                  | 3.84 | 0.97   | 0.03 |
> | DroneRun   | 0.77                  | 9.38 | 0.35   | 0.58 |
>
> The results, presented in the revised Table 7. This variant fails the safety constraint on every representative task, with normalized costs ranging from 1.11 to 9.38. This experiment provides definitive proof of our central claim: a cost classifier is an inherently unreliable mechanism for enforcing hard safety constraints, validating our design choice to use CFG for cost.
>
> ### W2:
> Thank you for this suggestion. We have added a detailed motivation and quantitative ablation for the SS-DA representation to the revised Appendix C.3.2.
>
> Our design is a deliberate trade-off:
> 1. Sparse-State (SS): To capture long temporal abstractions. This encourages broader exploration and more strategic high-level decision-making compared to traditional dense-state representations, which can be myopic.
> 2. Dense-Action (DA): We maintain a dense action sequence. This ensures the continuity of the generated trajectory.
>
> To quantitatively demonstrate the benefits of our chosen representation ($j=4$ by default) and its contribution to performance, we conducted an ablation study on the state sparsity interval $j$ in the Bullet-Safety-Gym environment.
> | Task | j=1 reward↑ | j=1 cost↓ | j=2 reward↑ | j=2 cost↓ | j=4 reward↑ | j=4 cost↓ | j=8 reward↑ | j=8 cost↓ | j=16 reward↑ | j=16 cost↓ |
> |---|---|---|---|---|---|---|---|---|---|---|
> | AntRun | 0.36 | 0.00 | 0.42 | 0.04 | 0.50 | 0.04 | 0.31 | 0.05 | 0.21 | 0.03 |
> | BallRun | 0.24 | 0.00 | 0.26 | 0.00 | 0.26 | 0.14 | 0.28 | 0.65 | 0.25 | 0.44 |
> | CarRun | 0.91 | 0.02 | 0.96 | 0.16 | 0.97 | 0.03 | 0.98 | 0.34 | 0.91 | 0.26 |
> | AntCircle | 0.13 | 0.79 | 0.24 | 0.94 | 0.20 | 0.98 | 0.15 | 0.85 | 0.03 | 0.96 |
> | BallCircle | 0.31 | 0.18 | 0.39 | 0.29 | 0.49 | 0.81 | 0.45 | 0.22 | 0.58 | 0.41 |
> | CarCircle | 0.32 | 0.88 | 0.47 | 0.90 | 0.40 | 0.90 | 0.54 | 0.93 | 0.54 | 1.37 |
> | **Avg** | **0.38** | **0.31** | **0.46** | **0.39** | **0.47** | **0.48** | **0.44** | **0.51** | **0.42** | **0.58** |
>
> Our ablation study on the state sparsity interval $j$ (revised Table 8) validates this. A fully dense representation ($j=1$) is the safest (0.31 avg. cost) but yields the lowest reward (0.38 avg. reward)1313. Our chosen default ($j=4$) achieves the highest average reward (0.47), demonstrating the optimal balance between long-horizon planning and safety.
>
> [1] Li, W., Wang, X., Jin, B., and Zha, H. Hierarchical diffusion for offline decision making. In International Conference on Machine Learning, pp. 20035–20064. PMLR, 2023.
>
> [2] Chen, C., Deng, F., Kawaguchi, K., Gulcehre, C., and Ahn, S. Simple hierarchical planning with diffusion. arXiv preprint arXiv:2401.02644, 2024.
>
> ### W3:
> 1. The primary goal of FTR is not to enforce the final cost limit $l$; that is, the explicit role of our cost-conditioned Classifier-Free Guidance (CFG). Instead, the goal of FTR is to purify the reward signal used to train the reward classifier ($p_{\phi}$).
> A key failure mode is when the reward classifier learns a spurious correlation that "high reward is associated with high cost". Our strict rule for the first $f$ steps is designed to break this specific correlation.
> Thus, it is precisely what enables our model to successfully decouple reward optimization from safety violations, allowing the CG to maximize rewards without compromising the safety enforced by the CFG.
> 2. We Do Not "Discard" Trajectories; We Re-Label Them. The reward classifier ($p_{\phi}$) also sees these trajectories. FTR simply re-labels their cumulative reward with a large penalty. For example: For example, in a trajectory (${s_0,s_1,..,s_n}$), $s_0$ and $s_1$ have cost, while $s_{n-1}$ and $s_{n}$ have high reward. We penalize (${s_0,s_1,..,s_n}$), but for trajectories like (${s_2,s_3,..,s_n, s_{n+1}, s_{n+2}}$), we can still learn high-value trajectory segments.

---

> ### Author Response · Authors · 2025-11-19
> **Weakness 4- Question 2**
>
> ### W4:
> It is entirely correct that formal Pareto optimality must be established on a per-task basis. Our intent with Figure 4 was not to claim formal per-task optimality, but to visualize the aggregate reward-cost trade-off across the entire DSRL benchmark.
>
> We have revised the language in the revised version to be more precise.
>
> 1. The Abstract now states that SDGD "extends the aggregate Pareto frontier... achieving a superior trade-off in the safe region" (revised from "sets a new Pareto-dominant... frontier" ).
> 2. The Figure 4 caption has been updated to clarify that SDGD "is the only method to achieve an average cost within this safe region, establishing a superior trade-off".
>
> ### W5:
> Thank you for noting this omission. We apologize for the oversight. All experiments were indeed run over 3 seeds.
>
> We have already included the standard deviation as error bars in our dual-guide ablation study (Figure 7) to demonstrate the robustness of our core design. For full transparency, the raw data (mean ± std) for that figure is provided below (data from Figure 7 ).
>
> | Environment | Task | w/o CFG (Reward) | w/o CFG (Cost) | w/o CG (Reward) | w/o CG (Cost) | SDGD (Ours) (Reward) | SDGD (Ours) (Cost) |
> |:---|:---|:---|:---|:---|:---|:---|:---|
> | Easy | Sparse | 1.00 ± 0.00 | 8.38 ± 0.18 | 0.57 ± 0.01 | 0.00 ± 0.00 | 0.62 ± 0.00 | 0.02 ± 0.00 |
> | | Mean | 1.00 ± 0.11 | 7.46 ± 0.71 | 0.56 ± 0.00 | 0.00 ± 0.00 | 0.60 ± 0.04 | 0.39 ± 0.41 |
> | | Dense | 1.00 ± 0.16 | 6.21 ± 0.92 | 0.68 ± 0.01 | 0.41 ± 0.06 | 0.71 ± 0.01 | 0.82 ± 0.05 |
> | Medium | Sparse | 0.17 ± 0.01 | 1.09 ± 0.02 | 0.16 ± 0.00 | 0.10 ± 0.00 | 0.17 ± 0.01 | 0.10 ± 0.02 |
> | | Mean | 0.25 ± 0.03 | 1.91 ± 0.33 | 0.17 ± 0.00 | 0.10 ± 0.00 | 0.19 ± 0.01 | 0.11 ± 0.00 |
> | | Dense | 0.18 ± 0.03 | 1.05 ± 0.28 | 0.18 ± 0.00 | 0.10 ± 0.00 | 0.20 ± 0.02 | 0.11 ± 0.01 |
> | Hard | Sparse | 0.44 ± 0.00 | 3.00 ± 0.08 | 0.42 ± 0.05 | 0.00 ± 0.04 | 0.44 ± 0.01 | 0.32 ± 0.04 |
> | | Mean | 0.44 ± 0.00 | 2.80 ± 0.13 | 0.38 ± 0.12 | 0.00 ± 0.05 | 0.45 ± 0.13 | 0.06 ± 0.06 |
> | | Dense | 0.44 ± 0.01 | 3.05 ± 0.18 | 0.35 ± 0.14 | 0.04 ± 0.00 | 0.44 ± 0.17 | 0.35 ± 0.01 |
>
> Furthermore, to address your concern about the main performance tables (e.g., Table 1), we provide a representative sample of results from the Safety-Gymnasium 'Velocity' tasks, comparing SDGD's stability against the baselines:
>
> | Task | CDT (Reward) | CDT (Cost) | TREBI (Reward) | TREBI (Cost) | CAPS (Reward) | CAPS (Cost) | CCAC (Reward) | CCAC (Cost) | SDGD (Reward) | SDGD (Cost) |
> | :--- | :--- | :--- | :--- | :--- | :--- | :--- | :--- | :--- | :--- | :--- |
> | AntVel | 0.89 ± 0.03 | 3.08 ± 0.60 | 0.31 ± 0.01 | 0.00 ± 0.00 | 0.89 ± 0.02 | 0.38 ± 0.22 | 0.32 ± 0.00 | 0.00 ± 0.00 | 0.90 ± 0.01 | 0.01 ± 0.00 |
> | HalfCheetahVel | 0.56 ± 0.05 | 1.90 ± 0.39 | 0.87 ± 0.02 | 0.23 ± 0.27 | 0.88 ± 0.01 | 0.31 ± 0.08 | 0.85 ± 0.03 | 0.92 ± 0.31 | 0.83 ± 0.01 | 0.00 ± 0.00 |
> | HopperVel | 0.17 ± 0.13 | 0.86 ± 0.17 | 0.08 ± 0.01 | 7.39 ± 4.97 | 0.17 ± 0.04 | 0.01 ± 0.05 | 0.11 ± 0.00 | 0.68 ± 0.04 | 0.63 ± 0.02 | 0.34 ± 0.04 |
> | SwimmerVel | 0.67 ± 0.32 | 1.47 ± 0.74 | 0.42 ± 0.09 | 1.31 ± 0.17 | 0.43 ± 0.18 | 7.01 ± 9.08 | 0.00 ± 0.00 | 2.44 ± 0.04 | 0.47 ± 0.01 | 0.15 ± 0.03 |
> | Walker2dVel | 0.74 ± 0.04 | 0.32 ± 0.27 | 0.10 ± 0.05 | 0.72 ± 0.29 | 0.8 ± 0.00 | 0.01 ± 0.05 | -0.01 ± 0.00 | 0.00 ± 0.00 | 0.10 ± 0.01 | 0.79 ± 0.20 |
>
> ### Q2:
> This is an excellent suggestion to demonstrate the robustness of our SOTA claim across different safety requirements.
>
> We have performed this analysis and added a new section, Appendix C.1, to the revised version. This appendix now contains:
> 1. Table 3: Complete per-task results for all adaptive models under the L=20 cost limit.
> 2. Table 4: Complete per-task results for all adaptive models under the L=30 cost limit.
>
> These results confirm that SDGD maintains its superior safety and performance. Under L=20, SDGD is safe in 37/38 tasks (achieving 13 SOTA rewards). Under L=30, SDGD is safe in 38/38 tasks (achieving 15 SOTA rewards). This provides strong empirical support for the robustness and superior adaptability of our method.

---

> > ### Comment · Reviewer_oXtw · 2025-11-27
> >
> > I sincerely thank the authors for the detailed response and additional experiments. The new ablations successfully address my primary concerns regarding the method's validity. I have raised my score to 6.
> >
> > However, issues regarding reproducibility and hyperparameter sensitivity still weaken the paper.
> > Given the offline setting where online tuning is impossible, I recommend demonstrating performance using fixed hyperparameters, at least per group of tasks, rather than fine-tuning for every individual task. This should serve as the main method, with the primary results and ablations based on it.

---

### Official Review · Reviewer_BTMc · 2025-11-07

**Soundness:** 3
**Presentation:** 4
**Contribution:** 4
**Rating:** 8
**Confidence:** 3

**Summary:**

The paper proposes SDGD (Safe Dual-Guide Diffuser) to target diffusion-based trajectory generation for offline safe RL. The key idea is to decouple safety and performance during the generation by using dual guidance mechanics: classifier-free guidance to condition on cost limits, and classifier guidance to steer toward high reward. It also explores a technique called Feasible Trajectory Re-labeling (FTR) to prevent reward gradients from pulling trajectories into unsafe regions and getting trained on preventing spurious high-reward/high-cost correlations. The overall method comes with strong theoretical guarantees on safety and optimality.

Experiments show that on the 38 DSRL tasks, SDGD reports outperformance over several baselines in safety and performance, is able to do almost all tasks safely, and achieves a pareto-optimal reward cost frontier. Authors also show that it adapts well to changing (even intra-episode) cost limits.

**Strengths:**

Although diffusion based and classifier guidance has been explored previously, the dual-guide approach is novel for OSRL, and addresses known limitations (unreliable cost classifiers, sequential errors) in prior works by conditioning costs directly.

The FTR heuristic also seems like a simple yet effective safety mechanism. It is
“the specific cost limit applied for the current generation does not predetermine the limits for subsequent generations” - this is a noteworthy strength of the framework. The strong performance in handling intra-episode varying constraints also demonstrates this amply.

The authors perform comprehensive evaluation with several baselines, and ablations (covering guidance, scale, FTR). The flow is logical and the writing is easy to understand.

The paper provides clear inequalities bounding both cost violation and reward loss, and links safety guarantees to dataset coverage. This is somewhat rarely seen in OSRL prior works.

**Weaknesses:**

Some theoretical assumptions  concentration bounds) may be standard but may not hold in less data regimes. Maybe I missed it, but it would be good to see some validation on how they hold in simulated datasets in the paper or some real world ones. Some small discussion around the bounds’ behavior when these break would also be insightful.

“we evaluate all methods under a strict, uniform absolute cost limit of 10 across all tasks” - the reasoning behind this is not clear. It may also hide cost scales issues across environments, and seems a bit overfit to the framework.

Relatively minor:
Variance over the evaluations/seeds can also be included with the results. The paper notes 2.1 Hz generation but compute/memory comparison would also be insightful. The framework and techniques like FTR is somewhat hyperparameter sensitive, which limits plug-and-play use.

**Questions:**

Can forms of FTR be applied to other baselines? It would be particularly interesting to see the performance if so.

In case of model failure (I believe in ~5% of the tasks), have the authors done any error analysis on the possible causes or common patterns, like poor dataset coverage, complexity etc.? Maybe it highlights more limitations of the model.

---

> ### Author Response · Authors · 2025-11-19
> **Rebuttal**
>
> Thank you for your thoughtful review and constructive feedback. We appreciate the opportunity to clarify these points and are encouraged by your positive assessment. We address your specific weaknesses and questions below.
> ## W1:
> We completely agree that the behavior of theoretical guarantees under limited data is of paramount importance. In direct response to your suggestion, we have conducted new experiments to empirically validate how the performance of SDGD and baseline methods degrades in low-data regimes, providing a clear view of the practical implications when the concentration assumptions are strained.
>
> We sub-sampled the offline datasets for two representative DSRL tasks (HalfCheetahVel and SwimmerVel) to 50% (p=0.5) and 25% (p=0.25) of their original size and re-trained all models. The results are summarized below:
>
>
> | Task                  | COptiDICE |        | CPQ    |        | CDT    |       | TREBI  |       | FISOR  |       | CAPS   |       | CCAC   |       | SDGD   |       |
> |-----------------------|-----------|--------|--------|--------|--------|-------|--------|-------|--------|-------|--------|-------|--------|-------|--------|-------|
> |                       | reward    | cost   | reward | cost   | reward | cost  | reward | cost  | reward | cost  | reward | cost  | reward | cost  | reward | cost  |
> | HalfCheetahVel-p=1    | 0.64      | 0.00   | 0.08   | 2.56   | 0.56   | 1.90  | 0.87   | 0.23  | 0.89   | 0.00  | 0.88   | 0.31  | 0.85   | 0.92  | 0.83   | 0.00  |
> | HalfCheetahVel-p=0.5  | 0.64      | 0.00   | 0.49   | 4.71   | 0.51   | 1.98  | 0.80   | 0.45  | 0.89   | 0.00  | 0.85   | 0.37  | 0.88   | 1.83  | 0.83   | 0.00  |
> | HalfCheetahVel-p=0.25 | 0.60      | 0.00   | 0.60   | 4.85   | 0.60   | 2.24  | 0.73   | 0.85  | 0.88   | 0.00  | 0.85   | 0.28  | 0.85   | 0.95  | 0.82   | 0.15  |
> | SwimmerVel-p=1        | 0.58      | 23.64  | 0.31   | 11.58  | 0.67   | 1.47  | 0.42   | 1.31  | -0.02  | 0.01  | 0.43   | 7.01  | 0.00   | 2.44  | 0.47   | 0.15  |
> | SwimmerVel-p=0.5      | 0.61      | 31.54  | 0.31   | 13.97  | 0.64   | 2.24  | 0.40   | 1.53  | 0.05   | 0.20  | 0.23   | 1.27  | 0.14   | 1.86  | 0.47   | 0.20  |
> | SwimmerVel-p=0.25     | 0.55      | 25.08  | 0.48   | 38.93  | 0.66   | 6.43  | 0.39   | 1.65  | 0.05   | 0.20  | 0.30   | 1.33  | 0.19   | 1.83  | 0.44   | 0.24  |
>
> 1. The results validate the expected behavior of the concentration bounds. As data diminishes (p decreases), the estimation error increases, leading to performance degradation for nearly all algorithms.
> 2. Crucially, our method, SDGD, exhibits remarkable robustness to this degradation. While its performance also reflects the increased uncertainty (e.g., a slight reward drop and cost increase in HalfCheetahVel-p=0.25), the magnitude of this decline is significantly smaller than that of most baselines.
>
> ## W2:
> We apologize for the lack of clarity. Our reasoning for choosing a uniform limit of 10 was not to overfit our framework but to establish a more rigorous and unified evaluation standard.
> To prove our method's adaptability and demonstrate that it was not "overfit" to this single value, we included extensive experiments at other cost limits.
>
> We have now included full benchmark results for all 38 tasks against all baselines for cost limit = 20 (Appendix C.1, Table 3) and cost limit = 30 (Appendix C.1, Table 4)
>
> These comprehensive results across three different cost limits (10, 20, and 30) confirm that our single SDGD model can robustly adapt to various safety requirements, consistently achieving a strong balance of safety and performance without retraining.
>
> ## W3:
> 1. All experiments in our paper were conducted over 3 seeds. We have already included the standard deviation as error bars in our core ablation study on the dual-guide components (Figure 7) to demonstrate the stability of our primary mechanism.
> 2. Taking nine tasks under Metadrive as an example, the state dimension is 261, while the action dimension is 2. Therefore, using the sparse-state dense-action trajectory representation method (with a default interval of 4 steps between states) can theoretically reduce memory requirements by nearly 4 times.
> 3. We acknowledge that the guidance scales ($s$ and $w$) are sensitive, which is a common and inherent characteristic of guided diffusion models. We found that a default setting of $s=0.02$ and $w=2.0$ provides a strong baseline for most environments. Some tasks do achieve better results with other values within our search space. Our complete codebase with specific settings will be made publicly available upon acceptance.

---

> > ### Author Response · Authors · 2025-11-19
> > **For Questions**
> >
> > ### Q1:
> > In principle, yes.
> > FTR is a dataset pre-processing technique that relabels the cumulative reward $R(\tau)$ for trajectories based on their initial feasibility.
> > Any baseline method that learns a reward model, a reward classifier, or even a Q-function (which implicitly models reward) from the offline data could potentially benefit from training on these relabeled rewards.
> >
> > ### Q2:
> > We thank the reviewer for their insightful question regarding the failure cases. As suggested, we have added a new detailed failure case analysis to Appendix C.4 (Failure Case Analysis).
> >
> > This new section provides a data-driven comparison between the failed DroneCircle task and the successful DroneRun task from the same environment.
> >
> > Our key findings are:
> > The failure is not due to a higher quantity of unsafe data.
> > The root cause is the qualitative structure of the DroneCircle dataset, which we term "temporal fragmentation." The trajectories consist of frequent, short-lived "hops" into and out of the unsafe region (e.g., mean violation length of only 12.1 steps, as shown in Figure 9).
> >
> > As visualized in our new PCA plot (Figure 10), this temporal fragmentation creates a spatially ambiguous, non-separable safety boundary, with safe (blue) and unsafe (red) states being highly intermingled.

---

> ### Comment · Reviewer_BTMc · 2025-11-23
>
> I sincerely thank the authors for the rebuttal. It significantly increased my understanding of the paper and continue to appreciate the merits of this work. I still share some concerns around reproducibility, sensitivity to task specific hyperparameters with other reviewers, so would like to keep my score.

---

### Author Response · Authors · 2025-12-02
**To the Area Chair (AC):**

We sincerely thank the Area Chair and all four reviewers for their time and expertise. We are encouraged that three of our four reviewers (BTMc, oXtw, Ea7w) now recommend acceptance, with scores of 8 (Accept), 6 (Accept), and 6 (Weak Accept), respectively.

The reviewers provided rigorous and challenging feedback, which spurred us to conduct a substantial number of new experiments during the rebuttal period to validate our claims. We believe these additions, now integrated into the revised paper (marked in blue), have significantly strengthened the submission.

Specifically, in direct response to reviewer feedback, we have added:
1. The "Swap Ablation" (New Appx. C.3.1, Table 7): At the request of oXtw, we performed the most critical ablation: inverting our model to use CG-for-Cost and CFG-for-Reward. The catastrophic failure of this variant (avg. cost 3.65) provides definitive empirical proof of our paper's core hypothesis: that cost classifiers are an unreliable mechanism for hard safety constraints.

2. Full Cost Limits L=20 and L=30 Benchmarks (New Appx. C.1, Tables 3-4): To address concerns from oXtw and Ea7w about the generality of our L=10 results, we have added complete 38-task benchmark tables for L=20 and L=30. These results confirm our single model remains SOTA, achieving safety in 37/38 (L=20) and 38/38 (L=30) tasks.

3. SS-DA Ablation (New Appx. C.3.2, Table 8): In response to oXtw, we added a quantitative ablation on our trajectory representation, validating that our chosen sparsity ($j=4$) achieves an optimal balance of high reward and safety.

4. FISOR Tuning Analysis (New Rebuttal Table for R4): To address Ea7w's crucial concern of an "unfair comparison," we followed their suggestion and tuned FISOR. We confirmed FISOR can be made safe, but this "eviscerated" its task performance (e.g., reward dropped from 0.58 to 0.07 on CarCircle2). This new result highlights the fundamental architectural superiority of our inference-time guidance over FISOR's train-time trade-off.

5. Failure Case Analysis (New Appx. C.4, Figs. 9-10): As BTMc requested, we added a deep-dive analysis of our model's (2/38) failures. The analysis (including new PCA plots) shows the failure is data-driven (ambiguous boundaries, temporal fragmentation) rather than a flaw in our method.

6. Low-Data Regime Experiments (New Rebuttal Table for BTMc): We added experiments showing SDGD's robustness as data is sub-sampled to 50% and 25%.

We are pleased that this new evidence successfully convinced oXtw, who raised their score to 6 (Accept), stating our new ablations "successfully address my primary concerns." This also resolved the "unfair comparison" concern from Ea7w, who maintained their 6 (Weak Accept), stating our replies "have addressed my concerns." BTMc (8, Accept) was also satisfied that our new experiments addressed their points.

We respectfully believe the 2 score from 2ZRQ is based on a single, persistent misunderstanding of our method's core mechanism, which R3 articulated in their final comment:
"A concern remains: the reward classifier guidance (CG) undermines the safety guarantee of the cost classifier-free guidance (CFG)... the generated trajectory can become unsafe."

We 100% agree with this premise; a naive CG would undermine safety. This is not a flaw in our method; this is the exact problem we explicitly identify and solve via our Feasible Trajectory Re-labeling (FTR) mechanism.
1. FTR "purifies" the reward signal before the Reward-CG is trained by breaking the spurious "high-reward = high-cost" correlation (Fig 3c).
2. Therefore, our Reward-CG learns not to conflict with the Cost-CFG, as it has no incentive to guide sampling toward high-cost regions.

We provided this clarification in our final response to 2ZRQ, pointing directly to the empirical evidence in our paper that resolves this exact concern: Figure 8.

The "w/o f" (no FTR) experiment in Figure 8 shows exactly the failure case R3 described: as reward guidance increases, the policy becomes unsafe. The "$f=1, 2, 4$" (FTR enabled) experiments show our solution: reward guidance can be aggressively increased, yet the policy consistently remains safe below the cost limit.

Crucially, due to the ICLR procedural freeze on Nov 27th, reviewers were unfortunately no longer able to edit their reviews or post further replies.
We therefore believe 2ZRQ did not have an opportunity to acknowledge this crucial, resolving evidence.

Given that Reviewer BTMc, oXtw, and Ea7w were all convinced by our new experiments and clarifications (leading to scores of 8, 6, and 6), we believe Reviewer 2ZRQ's core concern is similarly and decisively resolved by the data in Figure 8. Since the procedural freeze prevented 2ZRQ from updating their review, we respectfully ask the AC to act as the final arbiter on this point by personally examining the evidence in Figure 8, which we are confident resolves this misunderstanding.

Thank you for your time and consideration.

---

### Meta-Review · Area_Chair_SsXY · 2025-12-05

**Summary:**

The paper proposes SDGD (Safe Dual-Guide Diffuser), a diffusion-based offline safe RL method that decouples objectives: classifier guidance steers generation toward high reward, while classifier-free guidance conditions on target costs to enforce safety. It adds a feasible-trajectory relabeling scheme that penalizes unsafe prefixes so the reward signal doesn’t lure generation into high-cost regions. The method adapts to time-varying cost limits, and achieves good safety with strong returns on the DSRL benchmark.

The paper received mixed reviews, and some concerns of reviewers seem to still remain after the rebuttal process.

**Reviewer Concerns:**

Some concerns that I think are still not well-addressed:
- Doubts about the problem setting, and why existing methods cannot solve it (Reviewer 2ZRQ).
- Reproducibility and hyperparameter sensitivity (Reviewer oXtw, Eq7w, BTMc).
- Reward guidance could undermine the guarantee of the cost CFG (Reviewer 2ZRQ)

**Reviewer Scores:**

**Will increase:**
Reviewer oXtw mentioned that he/she will increase to 6, but still have concerns regarding hyperparameter sensitivity.

**Will not change:**
Reviewer Ea7w mentioned that he will maintain his/her score.
Reviewer 2ZRQ still have concerns regarding the paper, and might not change the score.

---

### Decision · Program_Chairs · 2026-01-26

Reject